# Recent Real-Time Aerial Object Detection Approaches, Performance, Optimization, and Efficient Design Trends for Onboard Performance: A Survey

**DOI:** 10.3390/s25247563

**Published:** 2025-12-12

**Authors:** Nadin Habash, Ahmad Abu Alqumsan, Tao Zhou

**Affiliations:** 1Institute for Intelligent Systems Research and Innovation, Deakin University, Waurn Ponds, VIC 3216, Australia; a.abualqumsan@deakin.edu.au (A.A.A.); tao.zhou@deakin.edu.au (T.Z.); 2The Department of Mechatronics Engineering, School of Engineering, The University of Jordan, Amman 11942, Jordan

**Keywords:** UAV, aerialobject detection, edge, real time, efficient, lightweight, onboard, optimization

## Abstract

The rising demand for real-time perception in aerial platforms has intensified the need for lightweight, hardware-efficient object detectors capable of reliable onboard operation. This survey provides a focused examination of real-time aerial object detection, emphasizing algorithms designed for edge devices and UAV onboard processors, where computation, memory, and power resources are severely constrained. We first review the major aerial and remote-sensing datasets and analyze the unique challenges they introduce, such as small objects, fine-grained variation, multiscale variation, and complex backgrounds, which directly shape detector design. Recent studies addressing these challenges are then grouped, covering advances in lightweight backbones, fine-grained feature representation, multi-scale fusion, and optimized Transformer modules adapted for embedded environments. The review further highlights hardware-aware optimization techniques, including quantization, pruning, and TensorRT acceleration, as well as emerging trends in automated NAS tailored to UAV constraints. We discuss the adaptation of large pretrained models, such as CLIP-based embeddings and compressed Transformers, to meet onboard real-time requirements. By unifying architectural strategies, model compression, and deployment-level optimization, this survey offers a comprehensive perspective on designing next-generation detectors that achieve both high accuracy and true real-time performance in aerial applications.

## 1. Introduction

A growing trend in the advancement of autonomous Unmanned Aerial Vehicles (UAVs) is their use for real-time mapping, alongside the implementation of deep learning techniques for the semantic analysis of data collected by UAVs [1]. Developing autonomous Unmanned Aerial Systems (UASs) primarily depends on their detection ability of static and moving objects [2], which makes UAV object detection essential during navigation for the aid of autonomous ground vehicle path planning [3,4]. UAVs have a broad range of applications that rely on object detection beyond navigation. They are employed in security and surveillance, including border patrol and the efficient monitoring of significant areas. UAVs provide valuable visual data to help responders plan rescues and locate survivors [3]. In the industry, UAVs play a vital role in inspecting infrastructure and monitoring environmental health. For example, UAVs are used for asphalt pavement cracking [5], wildlife conservation [6], and forest health checking (detecting dead trees or the incidence of disease) [7]. Similarly, UAVs are useful for mining exploration because they can provide relevant information on geological features, mineral deposits, and surface topography. In the same sense, UAVs are valuable tools in the construction industry. They help construction sites monitor safety regulations and compliance, as well as track construction progress.

RGB images are extensively used in aerial detection due to their rich colour and texture features. However, aerial detection poses various challenges. The distant areas of UAV aerial images frequently include numerous densely clustered small objects with few pixels of representation, making them difficult for standard detection algorithms to identify [8]. As a result, Information can be lost during the convolution process in feature extraction [9,10]. Previous studies have proposed several approaches to enhance the detection of small objects. One common strategy is replication augmentation, in which small objects are copied and randomly placed at different locations within the image, thereby increasing the number of small-object samples available for training [11]. Another approach involves scaling and splicing, in which larger objects in the original image are resized to generate additional small object instances. In addition, some methods focus on loss function adjustment, giving higher weight to the losses associated with small objects to encourage the model to pay more attention to them during training [11]. And some recent studies add an extra detection head to shallower layers for small object detection. However, improving the detection of small objects generally comes with an increase in model complexity and computational requirements, making it increasingly difficult to satisfy the requirements for real-time applications on memory- and processing-constrained platforms [12].

Fine-grained discrimination remains a major challenge in aerial detection, where visually similar classes often appear with only a few pixels and minimal texture. Although attention mechanisms, transformer-based feature alignment, and contrastive learning can significantly enhance fine-grained representations, feature map is another solution, in addition to spatial attention; these strategies typically increase computational burden and require more complex pipelines, limiting their suitability for real-time onboard deployment [13]. In low resolution, Spatially-adaptive Convolution SPDConv combined with Efficient Channel Attention ECA is used for fine-grained downsampling, and swin transformers for long-range dependancies [14]. This offers real-time performance but not on edge device.

High-resolution images facilitate the detection of small objects and the extraction of fine-grained features. However, increasing the resolution increases complexity. High-resolution feature extraction networks [15] for small object detection are complex; however, some emerging solutions include adding a super-resolution module that learns high-resolution features from low-resolution input, and this module is applied only during training to enhance detection while removed in inference for faster detection [16]. In [17] Fine-grained Information extraction module based on SPDConv with a multiscale feature fusion module based on BiFPN with skip connections to enhance fine-grained feature representation.

Another challenge is the scale and perspective differences: Objects may appear at different sizes and orientations depending on their distance from the camera and the angle of view. This challenge is addressed by using strategies such as Feature Pyramid Networks (FPNs), which have been widely applied to new detection tasks. These approaches advocate improved detection of small objects by leveraging features from multiple model layers. Nevertheless, they involve considerable computational requirements, potentially affecting real-time performance, especially on devices with limited resources [18].

Moreover, the orientation of objects presents another challenge [19]. Earlier datasets feature horizontal boundary boxes that are suitable for ground platforms. This can lead to inaccuracies in aerial detection, especially in detecting dense scenes. Addressing this issue depends on the algorithm; many studies adopt single-stage anchor-based algorithms with Horizontal Boundary Box (HBB) detections. Additionally, some studies employ Oriented Boundary Box (OBB) detections to handle objects at various angles. OBB detection can start with predefined anchors of different sizes and angles, or use horizontal anchors. After the initial detection, the model applies additional algorithms or layers to refine these anchors. This refinement process involves predicting offsets or adjustments to the position, size, and orientation of the anchors [20]. In anchor-free methods, the orientation is solved by different strategies, including regressing width, height, and angle at each center point using anchor-free feature maps [21]. Gaussian centerness and pseudo-domain angle encoding are used for OBB regression in [22], while ref. [23] uses channel expansion and dynamic label assignment is implemented to regress rotated boxes efficiently for real-time UAV imagery. These solutions increase the complexity of the models, affecting real-time inference.

In aerial remote sensing, a complex environment refers to a variety of scene conditions that make accurate object detection more difficult. Such conditions include background clutter, where targets are easily confused with surrounding structures, partial occlusions, in which objects are obscured by other elements in the scene, and changes in illumination, including shadows, bright sunlight, and low-light scenarios that alter visual appearance. Complex environments often contain unusual or previously unseen objects, which can significantly degrade object detection performance [19,24]. Context-Aware Compositional Networks address this challenge by modeling the relationships between object parts and their surrounding context, supplemented by additional contextual enhancements [25]. A solution to this includes self-attention in transformer models and dilated and cross-convolutions, which expand the receptive field but do not offer a lightweight solution for the process. Although methods such as attention mechanisms, Transformer-based feature alignment, and contrastive learning improve the representation of fine-grained local features, they are typically computationally demanding and rely on more complex processing pipelines [26].

Additionally, class imbalance in training datasets poses a problem. Some object classes are overrepresented, while others have limited samples, leading to models that perform well on common classes but poorly on rare ones. To mitigate this in aerial object detection, researchers adopt a variety of strategies, spanning data augmentation, loss design, feature representation, and learning paradigms. At the data level, techniques like K-means SMOTE generate synthetic minority class samples, while augmentation methods such as object copy paste and aggregated-mosaic increase the occurrence and contextual variation of rare objects [27]. For the algorithm loss, reweighting mechanisms, including Class Balanced Loss, based on the effective number of samples [28], focus training on difficult examples. For feature-level improvements, modules like Discriminative Feature Learning and Imbalanced Feature Semantic Enrichment [29], promote both semantic richness and better separation of rare classes. Finally, improved training protocols, such as duplicating rare-class objects (Auto Target Duplication) and ensuring that every augmentation patch contains at least one object (Assigned Stitch), help ensure that minority categories are sampled more effectively, thereby helping models learn from rare instances more reliably [30]. In [31], a hard chip mining strategy is designed to balance class distribution and produce challenging training examples. The approach begins by creating multi-scale image chips for training the detector. In parallel, object patches are extracted from the dataset to form an object pool, which is then used for data augmentation to alleviate class imbalance. The detector trained on this augmented data is subsequently run on the modified images to identify misclassified regions, from which hard chips are generated. The final detector is trained using a combination of regular chips and these hard examples. Altogether, these combined methods reduce sample scarcity, sharpen feature discrimination, and strengthen detection performance on long-tailed aerial datasets. Addressing these challenges is essential to improving the reliability and accuracy of object detection systems across diverse and complex settings.

Ultimately, real-time processing requirements for applications such as surveillance demand fast yet accurate object detection models, a difficult combination that researchers must balance to achieve both accuracy and speed [32]. Deep learning models for detecting small objects at low altitudes can be quite complex and computationally intensive. This complexity can be challenging to implement with edge computing systems that have limited processing and memory resources available [33].

For applications that demand faster detection, such as military operations and search-and-rescue missions, there is a need for onboard processing directly on the UAV. They make extensive use of object detection for secure navigation, to identify both dynamic and stationary objects, and for real-time decision making [3,4]. This is particularly relevant for applications such as path planning for unmanned autonomous ground vehicles, where UAVs assist with mapping and obstacle avoidance This enables real-time decision making without reliance on external networks, reducing communication delays and ensuring secure of information and rapid response in time sensitive situations.

Achieving real-time object detection processing on UAV platforms remains a significant challenge, particularly due to the limited computational resources. While improvements in GPU computing power have made object detection more accessible, optimizing performance while maintaining accuracy and efficiency remains an ongoing area of research. In object detection, Real-time processing initially relied on cloud-based computing, in which data is transmitted to remote servers for analysis. However, the need for lower latency and faster decision-making in various applications has led to the development of edge computing, making processing closer to the platform itself. In many applications, a hybrid approach, combining cloud and edge computing, is used, where edge processing is prioritized for speed-critical tasks, while cloud resources handle more complex computations and long-term data storage [34].

Real-time performance on UAVs with limited computational resources requires specialized algorithmic and architectural optimizations. Many approaches focus on designing efficient lightweight detection models to reduce computational overhead [33,35,36,37,38,39,40,41,42,43,44,45,46,47,48,49,50,51,52]. In addition to architectural improvements, various optimization techniques are applied to enhance efficiency further. One of these approaches is quantization, which reduces numerical precision, lowering computational complexity and memory usage, making the model more suitable for edge devices [21,45,51,53,54,55,56,57]. Also, Pruning is another optimization method that eliminates low-importance weights, reducing model size while maintaining performance [5,45,58,59,60,61,62]. In addition to knowledge distillation, which allows a smaller model to learn from a larger, more complex model, retaining essential knowledge while significantly reducing computational demands [63,64]. These optimization approaches, whether used separately or together, play a critical role in achieving real-time object detection on UAVs.

Several recent surveys have examined object detection from different perspectives. For example, ref. [65] provides a broad 20-year overview of object detection algorithms and their evolution, while ref. [66] systematically categorizes detectors into traditional and deep learning pipelines, covering one-stage, two-stage, transformer-based, and lightweight methods. Their analysis highlights the strengths and weaknesses of classical approaches, summarizes performance gains achieved through architectural design and feature learning, and identifies gaps related to lightweight models through comparative experiments and evaluation metrics. However, although lightweight networks are mentioned, these works do not explain how such models are integrated and developed specifically for aerial object detection, as they have different challenges compared to general natural scene detectors, nor do they address real-time processing, compression techniques, edge deployment requirements, or suitable algorithms for embedded platforms.

Only a limited number of studies focus directly on aerial detection. Among them, ref. [9] summarizes the main challenges and recent advances in deep learning for aerial imagery and reviews frequently used datasets; however, it lacks a deeper discussion on how to handle these challenges effectively and does not address lightweight or real-time methods. Similarly, ref. [67] reviews techniques for small object localization in aerial images, and [68] surveys object-oriented detection, but both remain limited in scope. Real-time detection is considered in only a few works; for instance, ref. [69] focuses on FPGA-based implementations but does not cover other commonly used devices, nor does it discuss the unique challenges of aerial imagery that increase detection complexity.

The review in [39] provides a detailed analysis of real-time processing strategies, algorithm speed, and sensor usage in UAV-based object recognition. However, most of these studies primarily report overall performance metrics without examining architectural design, the evolution of each detector component, or the requirements for real-time onboard processing. They discuss lightweight network architectures used as full networks, without discussing the developed modules from these networks, which are integrated into the general detector’s design and structure to enable aerial detection and achieve real-time performance. Furthermore, they lack a detailed examination of quantization strategies and pruning methods, which are not detailed in those reviews. Deployment frameworks, lightweight design principles, and emerging trends in convolution methods, edge-transformers, efficient attention mechanisms, head design, and end-to-end lightweight pipelines suitable for real-time and edge-level deployment. And a lack of more details about emerging trends, including the emerging edge transformers, hardware-aware NAS, adapting large vision-language models or CLIP embeddings for aerial detection, or integrating NAS and compression methods with these models to improve performance, and emerging trends in this area towards real-time and edge performance for such models.

Although several surveys have examined aerial object detection, see Table 1, they exhibit important limitations that restrict their relevance to real-time UAV deployment. Existing reviews focus primarily on high-level algorithmic families and overlook the end-to-end onboard pipeline, including preprocessing, postprocessing, quantization, pruning, TensorRT acceleration, and cloud–edge latency. They also provide limited coverage, hardware-aware design, trained quantization rather than post-training quantization, and structured pruning methods. Transformer refinements are now essential for achieving fast and accurate models on edge devices. Critically, earlier surveys rarely analyze Neural Architecture Search (NAS) frameworks tailored for UAV constraints or the emerging use of CLIP-style semantic embeddings for domain adaptation, despite their growing role in improving robustness and cross-scene generalization. Dataset discussions also lack emphasis on fine-grained benchmarks and high-resolution scenarios, small-object definitions, and specific datasets for small objects. Finally, most reviews fail to provide quantitative comparisons mAP, FPS, parameter count, on real embedded devices such as Jetson Nano, Xavier NX, Orin, NPUs, or Raspberry Pi, and tarde offs with typical use of detectors.

By these gaps, this review concentrates on the architectural components of modern detectors, along with hardware-oriented considerations and adaptive strategies essential for achieving real-time aerial object detection. Our survey integrates lightweight architectural advances with the practical constraints of embedded UAV platforms, connecting efficient backbone design, small-object enhancement, and multi-scale fusion with deployment strategies such as quantization post-training and trained quantization, pruning types, re-parameterization, TensorRT optimization, and federated edge learning. By incorporating recent progress in UAV-oriented NAS, we highlight how automated architecture exploration produces models that better balance accuracy and latency under strict power budgets, which can be further integrated with other methods, such as pruning for further efficiency. Additionally, we address the emerging role of CLIP-based semantic and visual embeddings in domain adaptation, a role that prior reviews do not discuss. Figure 1 gives insight into the review structure and topics.

The survey contributions are summarized below:Small object definition in aerial datasets and fine-grained datasets.Specify the real-time processing approaches and analyses covering performance and the platforms used, general detectors, typical applications for each model, how to modify them to mitigate aerial detection challenges, and keep efficient, with more focus on real-time studies with limited resources.Systematically review real-time aerial processing approaches, including platform-level constraints, performance analyses, and the typical adaptations required to modify general-purpose detectors for aerial challenges while maintaining efficiency on resource-limited hardware.Analyze lightweight design strategies across the detection pipeline, covering backbone, neck, and attention mechanisms, and new research for the developed RT-DETR design for edge deployment.Performance evaluation of recent edge research.Explain additional optimization techniques, such as pruning and quantization, with details about the new method, including quantization-aware training.Present emerging compression and hardware-aware optimization methods, including their integration with large vision–language models and multimodal distillation, to enable efficient deployment on UAV and edge devices.We identify the key limitations in current real-time aerial object detection research and discuss open challenges, offering insights and future research directions to advance onboard, real-time UAV perception.

The remaining is organized as follows. Section 2 presents the commonly used datasets referenced in the surveyed studies, highlighting their characteristics and relevance to aerial detection tasks. Section 3 reviews real-time processing techniques and summarizes the performance of existing edge and onboard systems. Section 4 explains general object detection paradigms and general lightweight networks. Section 5 This section reviews lightweight optimization strategies for aerial object detection, covering efficient backbone and detector design, model compression techniques, and emerging hardware-aware methods. It also highlights recent trends in integrating large language models with compressed architectures to enhance aerial detection under real-time and resource-constrained conditions. Section 6 presents limitations in the current research, open challenges, and future directions. Finally, Section 7 presents the overall conclusions of this survey and summarizes the key insights derived from the reviewed literature.

## 2. Datasets and Recent Real-Time Research Applications

Aerial and remote-sensing datasets are collected from platforms such as UAVs, manned aircraft, and satellites. Therefore, they differ from natural-scene datasets, including COCO and Pascal VOC, in several measurable ways. UAV imaging is commonly grouped into three altitude bands: close-range eye-level flights at 0–5 m, low-to-medium operations between 5 and 120 m, and high-altitude aerial imaging conducted at heights exceeding 120 m [9]. Aerial imaging also includes remote sensing images collected from multiple sensors and platforms (e.g., Google Earth), including satellite images with multiple resolutions [70].

Aerial Images and natural scene ground-level images do not use the same definition of object scale. In the natural scene images, where objects appear at consistent scales, the term small object refers to objects measuring fewer than 32 × 32 pixels [71], medium objects 32 × 32 to 96 × 96 and large objects above 96 × 96 [72], while in aerial and UAV imagery, the height of the horizontal bounding box, referred to as the pixel size, is used as the measure of object scale. Based on this criterion, instances in the dataset are categorized into three groups: small (10–50 pixels), medium (50–300 pixels), and large (greater than 300 pixels) [70].

In aerial scenes, Fine-grained discrimination refers to the challenge of distinguishing visually similar classes that often occupy only a few pixels and exhibit minimal texture. Although several studies attempt to address this problem, they generally rely on aerial datasets not explicitly designed for fine-grained detection [14,15,17,18,19]. Among these, only ref. [50] employs a dataset explicitly created for fine-grained tasks, with the Mar-20 dataset [73] and SeaDronesee [74] serve as a benchmark curated specifically for this purpose. These datasets serve different applications, as described in Table 2, which offers a comprehensive overview of datasets commonly employed in aerial object detection, illustrating their practical applications across domains and corresponding references.

These datasets’ images are captured by different satellite and UAV platforms, which are commonly classified into three types: fixed-wing, rotary-wing, and hybrid designs. Fixed-wing drones operate at higher altitudes, cover large areas quickly, and offer long endurance due to their simple structure and efficient gliding capability, though they require runways for takeoff. Rotary-wing UAVs can hover, take off, and land vertically, and fly at low altitudes to capture detailed, high-resolution data, making them well-suited for close-range inspection and sensing tasks. Hybrid platforms combine the strengths of both systems [75].

**Table 2 sensors-25-07563-t002:** Summary of commonly used datasets in aerial object detection, including statistics and key attributes.

Dataset	Description	Images	Instances/Objects	Classes	Size/Resolution	Annotation Type/Notes	References
VisDrone [76,77]	Drone-captured images and videos	263 video clips with 179,264 frames and additional 10,209 static images	540k	10	765 × 1360 to 1050 × 1400	HBB; high proportion of small, occluded and truncated objects	[11,12,33,34,35,36,39,41,44,45,46,47,51,52,57,61,62,78,79,80,81,82,83,84,85,86,87,88,89,90,91,92,93,94,95,96,97,98]
DOTA [70,99]	Aerial/satellite images	2806 images (v1)	188,282 instances (v1)	15	From 800 × 800 to about 4k × 4k	OBB; rotated and multi-scale objects	[14,38,40,82,84,100]
UAVDT [101]	UAV-based vehicle detection,	80k images (Video)	841.5k vehicles	4	1080 × 540	HBB; low-altitude, high density, small object	[79,87,88,95,102]
CARPK [103]	Parking lot vehicle counting	1448 images	89,777 vehicles	1	1280 × 720	HBB; small-to-medium vehicles	[6,91,104,105]
DroneVehicle [106]	RGB-Infrared vehicle detection	28,439 RGB-Infrared pairs	953,087 both modalities	5 Vehicles	840 × 712	OBB; multimodal RGB-IR	[40,46,85]
NWPU VHR-10 [107]	High-resolution remote sensing	715 from google earth images 85 images from Vaihingen data set	2.934k instances	10	from 533 × 597 px up to 1728 × 1028	HBB; planes, ships, vehicles	[46,105]
DIOR [108]	Optical remote sensing images	23,463 images	192,472 instances	20	800 × 800, 0.5–3 0 m	HBB; large-scale, cross-sensor variety	[14,33,84]
UA-DETRAC	Vehicle detection	10 h of video (140,000 frames)	1.21 million vehicles	4	960 × 540	HBB; traffic videos	[45]
UCAS-AOD	Aerial vehicle detection	1510 images	2500 instances	2	1000 × 1000	HBB	[84]
SODA-A [109]	Small object detection in aerial images	24,000 images	338,000 small instances	10	800 × 800	HBB; tiny objects	[38]
SEVE [110]	Small object detection	17,992 pairs of images and labels	-	10	1920 × 1080	HBB; Special vehicles in construction sites	[110]
FSD	small target scenes of fire and smoke	7534 images	N/A	N/A	N/A	HBB; 3 Fire hazard scenario, 3 non-hazard scenario	[42]
PVD	Photovolatic point defects (PDs) and line defects (LDs)	1581 images	2721	1	N/A	HBB	[79]
SAR-SD	Ship Detection dataset	1160 images	2456	1	resolutions (1 to 15 m).	HBB	[111]
AU-Air	Low altitude traffic survellance	32,823 images	132,034	8	1920 × 1080	HBB	[6]
Traffic-Net [112]	Traffic sign detection	4400 images	15,000 signs	1	512 × 512	HBB	[52]
Rail-FOD23	Railway infrastructure detection	2000 images	10,000 instances	1	512 × 512	HBB	[43]
Dead Trees	Forest dead tree detection	3000 images	15,000 instances	1	512 × 512	HBB	[7]
Mar-20	Remote-sensing image dataset for fine-grained military aircraft recognition	3842 images	22,341 instances	20	800 × 800	HBB; Fine grained	[50]
VEDAI [113]	Vehicle detection in parking lots	1210 images	3640 vehicles	9	(12.5 cm × 12.5 cm per pixel, 1024 × 1024	HBB, multimodal, visible and Near infrared	[114]
UAV Image	UAV-based detection	10,000–50,000 images	50,000+ instances	Various	1024 × 1024	HBB	[115]
PVEL-AD	Photovoltaic panel defect detection	5000 images	20,000 defects	1	512 × 512	HBB; defect-focused	[116]
RSOD	Remote sensing small object detection	886 images	5000 instances	4	600 × 600	HBB; small objects	[105]
SeaDronesSee [74]	maritime search and rescue	54,000	400,000	6	1280 × 960 to 5456 × 3632	HBB; 91% small objects, finegrained	[117]
AFOs	maritime search and rescue	3647 images	39,991	1	1280 × 720 to 3840 × 2160	HBB; small objects in open water	[117]


*Aerial Object Detection Applications*


### 2.1. Surveillance and General Object Detection

One of the most prominent application areas is surveillance and general object detection from UAVs, where datasets like VisDrone [5,46,57,93] are extensively utilized due to their rich annotations, high scene variability, and suitability for real-time, low-altitude aerial scenarios. Another dataset used in surveillance is CARPK [91], which focuses on UAV-based vehicle detection and parking lot vehicle counting. Typical platforms are DJI Mavic, Phantom series (3, 3A, 3SE, 3P, 4, 4A, 4P) (DJI, Shenzhen, China), used in the VisDrone dataset and DJI M200 (DJI, Shenzhen, China) used in the DroneVehicle dataset, both of which are rotary UAVs.

### 2.2. Environmental Monitoring

Also, another vital domain of application is environmental monitoring. For example, outfall structure detection is conducted with tailored datasets created for certain environmental compliance cases [118]. Wildlife detection, which typically requires efficient models on UAV platforms, has datasets available, like WAID and AU-AIR [6], with annotations on aerial imagery of wildlife capturing various animal species and natural environments. Emergency response systems, such as UAV fire monitoring, use thermal and RGB imagery during and after fire events that have been annotated for real-time detection [119]. In addition to applications like dead tree detection in forestry, it enables health monitoring of vegetation through spectral analysis.

### 2.3. Remote Sensing and High-Resolution Geospatial Object Detection

An emerging area of research is focused on remote sensing and high-resolution geospatial object detection. For example, multi-class object detection in satellite imagery relies on datasets like DOTA, DOTA-CD and AIR-CD [100], which allow for object detection in different orientations and with multi-scale sizes, in both complex urban and rural settings. These datasets are important for accommodating scale and rotation variability prevalent in remote sensing images. The AERO system [34] likewise demonstrates an AI-enabled UAV observational platform that is trained on private aerial data as well as the VisDrone dataset. AERO seamlessly provides high-level object detection and detailed tracking of detected objects across its UAV deployments in real-world scenarios. Lastly, methods developed for ultra-high-resolution (UHR) imagery have employed frame-tiling approaches in conjunction with parallel YOLO processing [120], allowing for real-time inference without sacrificing detection accuracy. Remote sensing platforms include satellites and airborne digital monitoring systems.

### 2.4. Industrial and Public Safety

The applications of real-time aerial object detection are crucial for industrial and public safety monitoring and inspection across a variety of industries. For example, in the coal mining industry, the CMUOD dataset [48] is designed to replicate the visual conditions found in underground mining environments, such as low-light conditions and small spaces, when detecting objects. In urban mobility and traffic analysis, MultEYE has been developed for vehicle detection and tracking and uses a custom-built UAV-based traffic monitoring dataset to allow for real-time monitoring of crowded streets and flow analysis [120]. When considering industrial infrastructure monitoring more broadly, railway infrastructure inspection represents another key application. The use of the Rail-FOD23 dataset [43] is an example of how inspection is performed using high-resolution aerial images annotated to detect faults and foreign object debris along railway tracks to support the operational safety of trains. Likewise, in the energy space, UAV-based systems for aerial defect detection in photovoltaic (PV) panels have been used, as in [116], and can be utilized to identify surface defects, shading, and damage across large solar farms with minimal human interaction during inspection.

### 2.5. Search and Rescue (SAR) Operations

The use of UAV-based object detection has also been explored in search and rescue (SAR) operations. These tasks generally relate to spotting humans or objects in rough terrain, and researchers have evaluated algorithms with datasets such as DIVERSE TERRAIN, RIT QUADRANGLE, and AUVSI SUAS [121] that represent various environmental settings, such as urban, rural, and wooded. At the same time, the demand for real-time pedestrian detection in videos from UAVs has resulted in the development of lightweight detection models, which have been evaluated on a custom UAV-based pedestrian dataset [56]. In addition to marins SAR operations [117]. In SAR missions, fixed-wing UAVs are used for their long-range search [122].

## 3. Real-Time Processing Platforms

Numerous real-time object detection algorithms have been executed on desktop GPUs [33,38,39,40,41,42,45,46,47,78,79,81,82,83,84,86,87,89,91,105,110,123]. But for real-world applications, cloud computing, edge processing, or embedded processing are the most common. While cloud computing may provide substantial computational resources, it introduces latency when running methods such as UAV-based object detection or autonomous navigation in real time. Some studies have proposed a hybrid cloud-edge processor to optimize speed and cost effectiveness [124]. Real-time onboard performance is crucial, yet only a few studies have achieved near-real-time performance (<30 FPS), and fewer have successfully implemented real-time onboard detection at speed (≥30) FPS [125]. The standard video processing benchmark of object detection is 30 frames per second [126]. However, other studies record lower detection rates that are less than this level [47]. Despite the ability of high-performance desktop graphics cards, such as the GeForce GTX1080 Ti, GeForce RTX3090, and GeForce RTX4080, to achieve real-time performance [127], other computational systems remain difficult to match. Maintaining similar speeds across diverse computing environments, such as edge devices, remains challenging due to the computational demands of these algorithms, often resulting in severe speed degradation [49,50,90,115]. Table 3, which shows the performance of recent real-time studies on the VisDrone dataset with corresponding desktop platforms used.

YOLOv8 models are evaluated across various versions, including YOLOv8n, YOLOv8s, and YOLOv5, on platforms such as NVIDIA GeForce RTX 3090 and NVIDIA GeForce RTX 4090. YOLOv8s achieves mAP50 of 42% at a speed of 126 FPS and an inference time of 7.8 ms evaluated on the VisDrone2019 dataset [123]. In contrast, YOLOv5 models (e.g., YOLOv5m and YOLOv5l) on NVIDIA GeForce RTX 3080Ti show similar performances but with varying speeds and inference times. YOLOv5 achieved 94.6% precision, with an inference speed of 17.8 FPS and an inference time of 70 ms on a TITAN RTX [84], while YOLOv8s achieved 45.8% precision with 77.6 FPS at 9.9 ms inference time on an A30 platform on the UAVDT dataset [119]. Other datasets, such as VisDrone2021-DET and DOTA, showcase similar trends, with the YOLO models offering varying trade-offs among precision, inference speed, and model size across different hardware platforms. In terms of real-time performance, YOLOv5s and YOLOv8s demonstrate competitive inference speeds, with YOLOv5s-obb showing higher mAP values and faster inference times on specific datasets such as DOTA and CARPK [86].

Despite these limitations, research on RGB-based object detection using edge computing has shown promising results on the NVIDIA Jetson family of devices, achieving speeds between 14 FPS and 47 FPS [6,39,49,50,51,57,102,115,119,128]. These findings highlight the potential of optimizing object detection models for real-time performance on resource-constrained platforms. Most studies report performance on desktop GPUs, whereas real-world applications employ three processing approaches: cloud, edge, and embedded computing. The following sections highlight key studies within each of these categories.

### 3.1. Cloud Computing

Cloud computing is a model in which end devices, such as mobile phones, autonomous vehicles, and sensors, are connected to central servers for data processing across a large network. Central servers provide computing, storage, and digital services to users through the internet. The main drawbacks of this method are the increasing volume of edge data (collected data from drones that is transferred to ground station for processing), which is restricted by network bandwidth, and the time required to transfer data to the server, along with the need for an internet connection [129]. To overcome these issues, edge computing is used. To evaluate the trade-offs between computational capacity and latency in edge and cloud settings, experiments in [130] were performed using the YOLOv8s (FP16) model on the Orin Nano. The assessment focused on round-trip time, inference latency, and communication delay to characterize real-time performance across both environments, see Table 4. Round-Trip Time (RTT) (ms) represents the total duration required for a complete inference loop, covering both network transmission delays and model processing time. It provides an overall evaluation of the system’s performance in real time.

The cloud latency shows a low value of 6.82 ms compared to the edge at 32.59 ms; however, the cloud communication latency reduces its efficiency by 341.41 ms compared to the edge, 2.5 ms only.

### 3.2. Edge AI Object Detection

Edge AI applies AI processing near end users at the network’s edge. In edge computing data processing is performed directly on the device or node where the data is generated [131,132]. Devices capable of supporting edge computing include System-on-Chip (SoC), Field-Programmable Gate Array (FPGA), Application-Specific Integrated Circuit (ASIC), Central Processing Unit (CPU), and Graphics Processing Unit (GPU). FPGAs have been initially viewed as one of the most promising platforms for deploying AI models due to their low power consumption. However, their limitations in inference speed and the lack of broad support for deep-learning frameworks make them less competitive today. ASICs can deliver excellent performance for specialized applications, but their long development time and high cost restrict their ability to keep pace with the fast evolution of object detection methods. In contrast, ongoing advances in semiconductor manufacturing have significantly improved the performance of GPUs, CPUs, and memory. These developments have expanded the edge-level computational resources available for neural networks, making the deployment of AI models on edge devices increasingly practical [11]. GPU-based edge platforms are closely linked to the development of deep learning and have become widely adopted for object detection due to their enhanced computational power. Typical computing platforms designed for embedded applications generally consume no more than 15W when operating under load [102]. In video detection, the NVIDIA Jetson family are frequently combined with portable devices to facilitate online detection [133,134]. Four types of edge GPU-based platforms are commonly used in the literature: NVIDIA Jetson Nano [135], NVIDIA Jetson TX2 [115], and NVIDIA Jetson Xavier NX [136,137] NVIDIA Jetson AGX Xavier [127]. It is worth mentioning that GPUs have been first introduced for real-time aerial detection in 2019 [138].

These embedded GPU platforms have higher energy efficiency compared to laptop and desktop GPUs [139], where NVIDIA Jetson TX2 and Jetson Xavier NX models are mostly used due to their moderate power consumption compared to the other series [140]. Few studies adopt a cloud-edge collaborative framework, in which the cloud provides scalability with virtually unlimited resources for large-scale data or complex tasks. At the same time, edge devices can independently scale using lightweight models optimized for specific UAV tasks. In [124], It integrates an Edge-Embedded Lightweight (E2L) object detection algorithm with an attention mechanism that helps a model focus on the most relevant parts of the input when making decisions, enabling efficient detection on edge devices without sacrificing accuracy. A fuzzy neural network-based decision-making mechanism dynamically allocates tasks between edge and cloud systems. Experimental results demonstrate that the proposed framework outperforms YOLOv4 in edge-side processing speed (NVIDIA Jetson Xavier NX) and provides better overall performance than traditional edge or cloud computing methods in both speed and accuracy.

### 3.3. Embedded-Onboard Object Detection

Embedded systems such as FPGA and Raspberry Pi 4B are used in a few studies research [141,142]. These systems are typically deployed directly on the UAV, using embedded or edge GPUs to perform all processing, including object detection, onboard the platform itself [138]. Table 5 shows the performance of onboard detection algorithms on edge GPUs and embedded devices.

#### Field-Programmable Gate Arrays (FPGAs) and Onboard Aerial Object Detection

Field-Programmable Gate Arrays (FPGAs) have gained increasing attention in aerial object detection because they provide true hardware-level parallelism and extremely low latency. Their large arrays of programmable logic units and DSP slices allow convolution, activation, and pooling operations to run concurrently, enabling fast per-frame processing even under tight compute budgets, which is a requirement typical of UAV-based video streams (Directory of Open Access Journals; Propulsion Tech Journal). Unlike fixed accelerators, FPGAs also offer full reconfigurability, allowing neural network operations or classical vision pipelines to be mapped directly into hardware. This flexibility supports aggressive quantization, pipelined dataflows, and resource-aware custom designs, particularly suitable for embedded and UAV platforms [69]. Another major advantage is energy efficiency. FPGA-based detectors generally consume far less power than GPU systems because of their optimized dataflow and parallel execution, which is essential for battery-powered aerial vehicles operating in resource-constrained environments.

Lightweight versions such as Tiny-YOLOv3, YOLOv11-Nano, and YOLOv11-S illustrate how reducing depth, channel width, and detection heads directly lowers Look-Up Table (LUT), A Look-Up Table (LUT)BRAM, and Digital Signal Processing Slice DSP consumption. Recent FPGA implementations of YOLOv11 use techniques such as depthwise-separable convolutions, CSP-based blocks, memory-efficient routing, loop tiling, and multi-PE parallelism to achieve real-time performance despite limited on-chip resources. Other optimization strategies, such as convolution lowering, increase PE utilization and reduce DRAM traffic by converting k × k kernels into expanded 1 × 1 operations. Deployment on specialized FPGA DPUs further boosts performance, as demonstrated by LCAM-YOLOX running at 195 FPS on ZCU102 and YOLOv5 achieving over 240 FPS on Versal platforms with minor layer modifications. Additional FPGA-friendly designs, such as Tiny DarkNet, also achieve high throughput and low power through fixed-point quantization and spatial parallelism. Comparisons with SSD reveal that although SSD can achieve high throughput, its deeper backbones and multi-scale feature processing require significantly more memory and power, making real-time FPGA deployment less common. Overall, YOLO-based models remain the preferred choice for embedded detection due to their predictable dataflow, lower resource demands, and suitability for highly parallel FPGA architectures, while the optimal FPGA device ultimately depends on whether the application prioritizes power, speed, or resource efficiency [143].

## 4. Aerial Real-Time Deep Learning Object Detection Algorithms

A typical object detector comprises two main components: a backbone for feature extraction and a head for making predictions about object categories and bounding boxes, see Figure 2 and Figure 3. The backbone, usually made up of convolutional layers, produces feature maps that capture key visual information such as edges, textures, and shapes from the input image. Common backbone architectures include the Visual Geometry Group (VGG), Residual Network (ResNet), its extended version ResNeXt, DenseNet, MobileNet, SqueezeNet, and ShuffleNet. Based on the configuration of the detection head, detectors are generally categorized as either two-stage or one-stage models [9]. In recent advancements, an intermediate component called the neck has been introduced to aggregate and enhance multi-scale feature maps from different backbone layers before passing them to the head.

### 4.1. Two-Stage and Single Stage Detectors

The following Two-stage algorithms, such as R-CNN, SPP-Net, Fast R-CNN, and Faster R-CNN, achieve high precision but at the cost of slower inference, making real-time detection challenging. In contrast, one-stage algorithms like SSD, YOLO, and RetinaNet prioritize speed, making them suitable for real-time applications, though they often exhibit lower accuracy compared to two-stage methods [9].

Whether a detector uses a single-stage or two-stage framework, it still must choose between two main ways of producing bounding boxes: anchor-based or anchor-free detection. This distinction is especially important in aerial scenarios because it directly affects efficiency, small-object performance, and suitability for onboard real-time UAV deployment. Anchor-based methods rely on predefined bounding boxes during training to predict objects, whereas anchor-free approaches eliminate the need for predefined anchor boxes and directly predict. object locations from feature maps, offering greater flexibility and reduced computational overhead, see Figure 4.

#### 4.1.1. Anchor Based Methods

In many object detection models, the image is split into a grid, and predefined anchor boxes of various sizes and shapes are placed at each grid cell. These anchors guide the model in predicting different objects in the image by providing a set of initial guesses for where objects could be. The model then learns to adjust the position, size, and shape of these anchor boxes to better match the actual objects, refining them into the final predicted bounding boxes. A process called Non-Maximum Suppression (NMS) is used to remove overlapping boxes, keeping only the best one. The model uses Intersection over Union (IoU), a metric that measures the overlap between the detected bounding box and the ground truth, and confidence scores to decide which boxes to keep. The confidence score indicates how sure the model is that a box contains an object and what type of object it is. This process ensures accurate and non-redundant detections. A prediction is regarded as a true positive if the Intersection over Union (IoU) between the prediction and its closest ground-truth annotation exceeds 0.5. In the post-processing step called Non-Maximum Suppression (NMS), these predictions are filtered to result in one bounding box for each object [146].

While these methods are effective, they present several challenges, including false positives, difficulties with varying aspect ratios, and high computational costs [147]. The reliance on anchor boxes increases redundancy and computational overhead, complicating Non-Maximum Suppression (NMS), a technique that removes duplicate detections by retaining only the bounding box with the highest confidence score [146]. Additionally, anchor-based methods often suffer from poor generalization across different object shapes and create an imbalance between positive and negative samples, which complicates training, particularly for edge AI systems [22,148].

Within anchor-based detection, different bounding box strategies exist. The Horizontal Bounding Box (HBB) approach is computationally efficient and offers faster inference due to the lower number of anchors required. However, it frequently produces false positives, especially in dense scenes, where it may fail to distinguish among multiple objects or include non-object regions within the bounding box. In contrast, the Oriented Bounding Box (OBB) approach enhances detection accuracy by incorporating angle parameters, making it particularly beneficial for objects with large aspect ratios, such as vehicles and ships see Figure 5. Despite this advantage, OBB-based methods significantly increase computational complexity, requiring approximately six times as many anchors as HBB, making them considerably slower [149].

For high recall rates, anchor-based detectors must densely place a large number of anchor boxes in the input image, such as over 180,000 anchor boxes in FPN networks for an image with a shorter side of 800 pixels [145]. Recall measures how well the model finds all the relevant objects in an image. It is the ratio between the number of actual objects that the model successfully detects-true positives-and the total number of actual objects, which is the sum of true positives and missed objects. A high recall means the model detects most of the objects but might include some false positives. Balancing recall with precision, which measures the accuracy of the detections, is essential for overall performance [151]. During training, the majority of these anchor boxes are marked as negative samples. The large quantity of negative samples exacerbates the imbalance between positive and negative samples in training. In addition, it can lead to potential latency bottlenecks when transferring predictions between devices in edge AI systems [152]. To address this issue, anchor-free methods have emerged.

#### 4.1.2. Anchor Free Methods

Anchor-free detection eliminates the need for predefined anchor boxes and directly predicts object locations from feature maps. This approach improves adaptability to a range of object sizes and aspect ratios while significantly reducing computational complexity. Only a small number of anchor-free detectors are built on two-stage frameworks [21]. RepPoints [153] is a representative example that replaces anchors with learnable point sets that capture object extent and key semantic regions. SRAF-Net [154] also adopts an anchor-free strategy but embeds it within a Faster R-CNN structure, increasing complexity. Similarly, AOPG [152] introduces oriented proposals via feature alignment modules based on Faster R-CNN. Although these methods improve localization, especially for rotated objects, they remain computationally demanding and are not suitable for real-time UAV deployment. Popular anchor-free methods such as CenterNet [145], CornerNet [155], and FCOSR [21] adopt one-stage methods with different strategies, including predicting object centers, corners, or bounding box distances. Thereby avoiding the complexities associated with anchor box design and tuning [145]. These methods provide greater flexibility and efficiency in object detection by avoiding the constraints imposed by anchor boxes.

### 4.2. Lightweight Networks

In object detection models, the backbone is a critical component in lightweight design, often responsible for more than 50% of the total computational load during inference [156]. Many standard networks are too computationally heavy for real-time UAV or embedded applications, motivating the adoption of lightweight backbones in remote sensing image analysis [50]. Table 6 summarizes these backbones, highlighting their key convolution methods and structural strategies for efficient feature extraction.

The evolution of lightweight backbones begins with SqueezeNet (2016) [157], which introduces the Fire module to replace standard 3 × 3 convolutions with 1 × 1 kernels, reducing parameters while maintaining channel flexibility. MobileNetV1 (2017) [158] adopts depthwise separable convolutions, significantly improving computational efficiency compared to standard VGG-style convolutions. Its successor, MobileNetV2 (2018) [159], integrates residual connections and linear bottlenecks, combining expansion and compression of feature dimensions to enhance feature extraction.

ShuffleNet (2018) [160] focuses on channel-level operations, using pointwise group convolutions and channel shuffle to maintain information flow across groups. ShuffleNetV2 (2018) [161] refines this design by splitting the input channels into two branches, processing one while leaving the other unchanged, followed by concatenation and channel shuffle, thereby balancing computational cost and simplicity.

GhostNet (2020) [164] emphasizes redundancy reduction through cost-efficient linear operations to generate extra feature maps, and GhostNetV2 (2022) [166] extends this with Dynamic Feature Convolution (DFC) attention for better local-global feature extraction in mobile-friendly applications.

Finally, FasterNet (2023) [165] addresses inefficiencies in depthwise convolutions using Partial Convolution (PConv) to limit memory access and redundant computations. This enables FasterNet to achieve faster inference across GPU, CPU, and ARM platforms while maintaining competitive accuracy. For instance, FasterNet-T0 surpasses MobileViT-XXS with 2.8× faster inference on GPU, 3.3× on CPU, and 2.4× on ARM, while improving ImageNet-1k top-1 accuracy by 2.9%. Moreover, the FasterNet-L variant achieves an 83.5% accuracy, comparable to Swin-B, while delivering 36% higher GPU throughput and reducing CPU compute time by 37% [165].

### 4.3. Neck Network

The neck in object detection models acts as a bridge between the backbone and the head, aggregating and refining multiscale features for final prediction. Early structures such as the Feature Pyramid Network (FPN) [167], enriched shallow features with deeper semantic information, but its top–down design struggled to preserve spatial detail. PANet [168], adopted in [43,47], addressed this by adding a bottom–up pathway to strengthen shallow feature representation. EfficientDet later introduced the Bi-Directional FPN (Bi-FPN) [169], used in [86], which employs weighted top–down and bottom-up fusion for improved efficiency. Fusion in FPN-style structures typically relies on element-wise addition for efficiency [167], although concatenation is sometimes used when a richer representation is required. Neural Architecture Search (NAS) further automated feature pyramid design in NAS-FPN [170], Figure 6 shows the general neck structures.

### 4.4. Attention Modules

Attention modules play a critical role in enhancing feature representations by directing the model’s attention to the most informative regions of an image. In computer vision, attention mechanisms are predominantly categorized into two types: channel attention and spatial attention. Both types enhance original feature representations by aggregating information across positions, though they differ in their strategies, transformations, and strengthening functions [171]. Fundamental attention mechanisms and their enhanced variants have been employed throughout the research. Table 7 shows these basic mechanisms and their key features and limitations.

When comparing attention modules in terms of computational overhead, ECA attention introduces less overhead than CBAM and SE. At the same time, SA has a similar overhead to ECA. The Triplet Attention mechanism also maintains lower overhead compared to both CBAM and SE. Similarly, NAM (Normalization-based Attention Module) and SimAM (Simple Attention Module) are also lightweight alternatives, with SimAM being parameter-free and NAM maintaining low complexity, both significantly lighter than the mentioned methods.

The Squeeze-and-Excitation (SE-2018) module [172] effectively strengthens channel-wise feature encoding but does not account for spatial dependencies.

To address this limitation, modules such as the Convolutional Block Attention Module (CBAM-2018) [173], Bottleneck Attention Module (BAM-2018) [174], sequentially derive attention map in two dimensions, channel and spatial dimension; however, they attempt to utilize positional information by decreasing the channel dimension and then generating spatial attention by convolutions, but convolutions are limited to capturing local relationships and struggle to model long range, which are crucial for vision tasks.

Global context network (GCNet-2019) [175] incorporates spatial attention, though their reliance on convolutional operations constrains their capacity to model long-range dependencies. However, these integrated CBAM–GCNet designs often face challenges, including slow convergence and increased computational complexity.

Efficient Channel Attention (ECA-Net) [176] streamlines the channel attention mechanism of the SE block by employing a 1D convolution to compute channel weights more efficiently. Similarly, Spatial Group-wise Enhance (SGE) [177] divides the channel dimension into multiple sub-groups, each intended to capture distinct semantic representations, and applies spatial attention within each group using attention masks that scale feature vectors across spatial locations, adding minimal computational overhead and virtually no additional parameters. Despite these innovations, such models often underutilize the joint relationship between spatial and channel attention. Moreover, they typically do not incorporate identity-mapping branches, thereby limiting their overall efficiency and effectiveness.

Coordinate Attention (CA) [178] introduces positional encoding tailored for lightweight networks, improving localization while maintaining efficiency. Meanwhile, the Normalization-based Attention Module (NAM-2021) [179] selectively suppresses less relevant features, contributing to computational efficiency. Triple Attention [180] achieves a balance between cost and performance by modeling both spatial and channel-wise interactions without significant overhead. While transformer-based self-attention mechanisms offer strong global feature modeling capabilities, their high computational cost poses challenges for real-time applications on resource-constrained UAV platforms.

The Shuffle Attention (SA) [181] module provides an efficient solution by integrating two types of attention mechanisms through the use of Shuffle Units. It begins by dividing the channel dimension into several sub-features, which are then processed in parallel. Each sub-feature is passed through a Shuffle Unit to capture dependencies across both spatial and channel dimensions. Finally, the outputs from all sub-features are combined, and a channel shuffle operation is applied to facilitate information exchange among them.

Simple Attention Module (SimAM) [182] offers a unique approach compared to traditional channel-wise or spatial-wise attention mechanisms by directly generating 3D attention weights for each neuron in a feature map—without introducing additional parameters. SimAM formulates an energy function to assess neuron importance and derives a fast, closed-form solution that can be implemented efficiently. Its simplicity avoids the need for complex structural design or tuning. Recent research has demonstrated the strategic integration of attention modules across different stages of detection architectures.

### 4.5. Real-Time Aerial Object Detectors

All algorithms mentioned previously in Section 4 are general-purpose detectors, mainly trained and evaluated on natural-scene datasets. Because aerial imagery has distinct properties (e.g., small targets, dense object distributions, complex backgrounds), it faces specific challenges that require dedicated treatment. The majority of recent studies reviewed in this survey primarily adopt YOLO detectors, particularly YOLOv8, followed by YOLOv5, and then YOLOv7. New versions of YOLO have been released: YOLOv11 and YOLOv12, and YOLO-Gold. A summary of the specifications for each YOLO version and its typical use is presented in Table 8.

Most aerial object detection studies develop YOLO algorithms to meet the requirements of aerial challenges or to make onboard detection more efficient. The new YOLO models (YOLOv5, YOLOv8) come in different sizes, each optimized for specific use cases, balancing speed, accuracy, and computational efficiency. The smallest version, YOLO nano version (YOLOvn) is designed for extreme efficiency, making it suitable for edge AI applications and low-power devices where real-time processing is essential with minimal hardware requirements used for lightweight aerial detection as in [35,36,39,48,50,51,89,90,116]. YOLO-Small (YOLOvs) provides a balance between speed and accuracy, making it an ideal choice for real-time applications on embedded systems and mobile devices like in [7,12,33,37,43,48,51,81,82,83,87,118]. YOLO-Medium (YOLOvm) provides a trade-off between computational cost and detection performance. YOLO-M is well-suited for applications that require improved accuracy while maintaining reasonable inference speeds. YOLO-Large (YOLOvl) is more performant, with more parameters and higher accuracy, but has higher computational requirements. YOLO-X-Large (YOLOvx) is more accurate but more rigorous in development, and most often operates in a high-performance computing environment where real-time requirements are less important [199].

Table 9 summarizes the developed aerial studies along with the basic models they employ. These models have been adapted to suit various aerial detection scenarios and different application requirements, with real-time performance on desktop GPUs.

As shown in Table 9, for real-time applications, the smaller versions, such as YOLOvn and YOLOvs, are preferred because they achieve high-speed inference with lower latency, making them well-suited for tasks such as autonomous navigation, robotics, and surveillance. The ability to select a model based on the specific needs of an application allows YOLO to remain a versatile and widely used object detection framework in both edge and cloud-based AI systems, mainly the small and Nano ones. The performance evaluation of the lightweight detectors, YOLOv9t, YOLOv7tiny, YOLOv10n, and YOLOv10s has been conducted in [201], This study evaluated YOLO models on a Jetson Xavier-equipped drone, emphasizing speed, accuracy, and resource efficiency under TinyML principles. Key metrics included inference time, mAP, GPU usage, and power consumption. YOLOv10n achieved the best balance (10.34 ms, 0.657 mAP), while YOLOv9-tiny also performed well (13.55 ms, 0.688 mAP) with slightly higher resource use. YOLOv10s achieved the highest accuracy (0.77 mAP) but demanded more power and GPU resources, favoring accuracy scenarios. However, fewer approaches apply another detector, as in [119], in which SSD detector is used.

To achieve real time time performance diffrent methods have been adapted targeting the design of lightweight models aiming to reduce size and computational complexity [163,203,204], most of them achieve this by replacing the backbone with lightweight networks [119], or by changing some module in the network [7,33,36,38,40,43,47,61,62,78,79,81,82,83,84,86,89,104,110,111,200], with fewer studies have explored other methods such as parameter pruning [57,62], quantization [5,7,39,42,46,49,56,57,115] and Knowledge distillation [63,64]. The one-stage object detector comprises three main components: The backbone for feature extraction, the neck for multiscale feature fusion, and the head, which locates objects in images and assigns classes. Various optimization and design modifications have been applied to enhance the efficiency and speed of models, with the majority of these changes focused on the backbone [7,33,38,43,81,84,110,111] and neck structure [89,104]. Fewer studies modify the head [40,91]. Studies that change backbone and neck [36,40,47,61,62,78,79,81,82,83,86,200].

The Refined Anchor-Free Rotated YOLOX detector (R2YOLOX) [205], which is an enhanced version of YOLOX [188] for aerial imagery, is an example of an anchor-free YOLO version in addition to YOLOv8 and later versions, which have also adopted an anchor-free approach and been used in many research, further advancing real-time object detection performance [35,48,92]. In recent real-time research, various algorithms have been used for aerial object detection.

## 5. Optimization Methods

Aerial object detection systems deployed on UAV platforms face resource constraints in terms of computation, energy, and latency, making optimization a core requirement rather than an optional enhancement. Unlike ground-based detectors that can rely on powerful servers or desktop GPUs, onboard aerial systems must operate under limited computational resources while still providing real-time, reliable detection. Therefore, optimization methods in the literature aim to either reduce the computational burden, enhance detection accuracy under resource limitations, or balance both goals to meet real-time performance requirements. In [206], Model compression for object detection is generally categorized into five main approaches: lightweight network design, pruning, quantization, knowledge distillation, and neural architecture search (NAS), which are described in detail in the following subsections.

### 5.1. Lightweight Design for Real-Time Aerial Detection

The following subsections provide a detailed analysis of some studies, highlighting their contributions to onboard algorithm design and optimization. The following explains the design of the lightweight backbone, neck, and head.

#### 5.1.1. Lightweight Backbone Networks

Since the backbone is the most computationally intensive component, most lightweight models focus on designing efficient backbone architectures, following optimization strategies commonly adopted in aerial detection algorithms.

A.Convolution-Based Lightweight Designs

In real-time aerial detection, many studies replace the original backbone with lightweight architectures to improve efficiency under embedded constraints, as demonstrated in [40,41]. Recent backbone design trends focus on replacing standard convolutions with more efficient operators, particularly in YOLO-based models such as YOLOv5 and YOLOv8. Common approaches include Depthwise Separable Convolution (DWSeparableConv) [7,47,86], often coupled with channel shuffle operations [36,44], followed by broader adoption of Ghost Convolution [39,110], Partial Convolution (PConv) [38,43], and MobileNet Bottleneck Convolution (MBConv) [81,200]. Some models integrate hybrid dilated convolutions with partial convolutions to enrich receptive fields [38], while others employ advanced variants such as Omni-dimensional Dynamic Convolution (ODConv) [111] to introduce input-adaptive filtering.

Since the backbone is the most computationally demanding component of the YOLO architecture, it plays a central role in extracting multi-scale visual features before they are passed to the neck for further aggregation [207]. Therefore, optimizing the backbone directly affects both accuracy and inference speed. As illustrated in Table 10, which summarizes works achieving real-time performance on Jetson GPUs and embedded platforms, different studies target distinct architectural sections; however, nearly 85% apply lightweight modifications to the backbone, underscoring its dominant contribution to total computational cost.

Lightweight aerial detection backbones thus integrate a spectrum of strategies with efficient convolutions, compact attention modules, reparameterized multi-branch structures, and lightweight downsampling and fusion blocks to enable high-speed onboard inference.

A major category of these methods is convolution-based lightweight design, where computationally expensive operations are replaced with more efficient alternatives. ShuffleNetV2 is widely adopted in this context: YOLOv6 combines it with the Multi-Scale Dilated Attention (MSDA) module to enhance multi-scale processing [97]. At the same time, YOLOv5n incorporates ShuffleNetV2 together with Coordinate Attention to improve spatial sensitivity [50]. Rep-ShuffleNet continues this trend by removing the traditional 1 × 1 convolution used for dimensional alignment and introducing a depthwise convolution block (DwCB) within a dual-branch structure during training [48]. Ghost-based techniques are frequently used to reduce redundancy, as seen in WILD-YOLO, which replaces multiple YOLOv7 backbone modules with GhostConv [6]; MSGD-YOLO, which integrates Ghost and dynamic convolution for adaptive feature extraction [35]; and the MGC module, which merges MaxPooling, GhostConv, and PConv to minimize parameters [125]. A streamlined YOLOv8s variant similarly replaces parts of the C2f block with GhostBlockV2, reducing complexity while preserving representational capacity [90]. Complementing these approaches, the IRFM module combines multi-scale dilated convolutions with trainable weighted fusion to enhance contextual representation, especially for small-object detection in cluttered scenes [5].

B.Attention-Enhanced Lightweight Modules

These methods maintain representational quality by integrating efficient channel- or coordinate-attention mechanisms. Coordinate Attention (CA) has been incorporated into the YOLOv5n backbone alongside ShuffleNetV2 to enhance spatial and orientation awareness [50]. Similarly, Rep-ShuffleNet integrates Efficient Channel Attention (ECANet), which employs adaptive k-nearest neighbor based kernel selection and a refined shortcut design to strengthen channel-wise feature interactions [48]. Beyond these explicit attention mechanisms, other models adopt attention implicitly. MSGD-YOLO utilizes dynamic convolution, which inherently provides input-adaptive attention [35], whereas the IRFM module introduces a trainable weighted fusion strategy that acts as a soft attention mechanism to emphasize multi-scale contextual responses [5].

C.Reparameterization and Structural Re-Design

These approaches improve training expressiveness through multi-branch structures, then collapse them into efficient single-path inference architectures. RepVGG introduces a structural re-parameterization strategy in which multi-branch convolutional blocks used during training are merged into a single equivalent convolution at inference, enabling the model to maintain accuracy while substantially reducing MACs, an approach that forms the basis of the structural design used in Rep-ShuffleNet [48]. Building on this concept, Rep-ShuffleNet adopts a dual-branch architecture composed of a depthwise convolution branch (DwCB) and a shortcut pathway during training, which is then collapsed into a simplified structure for efficient inference [48]. Likewise, RTD-Net employs a Lightweight Feature Module (LFM) that distributes computation across homogeneous multi-branch pathways, reducing redundancy while preserving strong representational power [115].

D.Lightweight Pooling, Downsampling and Feature Fusion Modules

These modules enhance feature hierarchy while reducing interpolation or element-wise overhead. Lightweight downsampling and fusion strategies are also employed to strengthen feature hierarchies with minimal computational overhead. The MGC module integrates MaxPooling with GhostConv to achieve efficient downsampling through a combination of pooling and lightweight convolution [125], while the DSDM block in Spatial Pyramid SOD-YOLO enhances deep–shallow hierarchical representation using a compact downsampling structure [46]. Additionally, the IRFM module incorporates a trainable weighted multi-scale fusion mechanism applied over dilated convolutions, improving contextual encoding and boosting small-object detection performance in complex environments [5].

#### 5.1.2. Efficient Neck Networks

A.Enhancements to Classical FPN Structures for UAV Detection

In UAV scenarios, where objects are small and spatially sparse, enhanced feature pyramids are widely adopted. Dense-FPN (D-FPN) [104], used in YOLOv8, improves shallow-feature extraction by repeatedly integrating shallow and deep features using downsampling, upsampling, and convolution. The SPPF module enhances multiscale perception by extracting deep semantic context before merging with shallow layers, while the Dense Attention Layer (DAL) strengthens channel- and spatial-wise focusing via average, max, and stochastic pooling.

YOLOv5’s neck is modified in [79] to include a Bidirectional FPN (BDFPN), which expands the FPN hierarchy and introduces skip connections for improved multi-scale fusion, particularly valuable in dense UAV scenes with large object-scale variation.

YOLOv8 adopts a lighter PAN-FPN variant that removes post-upsampling convolutions to reduce computational load. However, this limits small-object detection due to reduced high-resolution feature fusion. To improve this, [123] presents BSSI-FPN, extending the pyramid upward to preserve spatial detail and adding a micro-object detection head. Additional downsampling blocks between the backbone and the neck further enhance semantic–spatial integration.

B.Attention- and Module-Based Feature Fusion Enhancements

Several designs enhance feature fusion via attention or cross-scale interactions. In [89], a PA-FPN is augmented with the Symmetric C2f (SCF) module for deep feature refinement and the Efficient Multiscale Attention (EMA) module for enhanced spatial–channel coupling. Its Feature Fusion (FF) block improves texture–semantic blending. Similarly, [12] integrates the AMCC module into PA-FPN to reduce redundancy and preserve small-object details via channel spatial attention.

The MLFF module in [62] merges shallow, middle, and deep features of identical spatial size but different channels to reduce computation while preserving multi-scale richness. The Scale Compensation Feature Pyramid Network (SCFPN) in [78] replaces the large-object detection layer with an ultra-small object layer to better target fine-scale features. Additional improvements include the C2f-EMBC module in [81], which replaces bottlenecks with EMBConv and adds SE-like attention for memory-efficient enhancement, and SCFPN’s adaptation in [82], which further strengthens the merging of low-level spatial cues with high-level semantics. Cross-layer and weighted fusion improvements also exist. CWFF [86] introduces cross-layer weighted integration while adding a high-resolution P2 layer to enhance micro-object detection.

C.Lightweight Neck Variants for Embedded and UAV Platforms

To reduce computation on embedded platforms, Slim-FPN [36] replaces YOLOv5-N’s neck with depthwise separable designs (DSC, DSSconv, DSSCSP, SCSAconv, SCSACSP), combining efficient convolution and attention for lightweight fusion. In [89], the EMA and FF modules capture cross-scale dependencies for UAV imagery, while the SCF module optimizes bottleneck placement. The CSPPartialStage from [38] further strengthens spatial information aggregation in complex aerial contexts.

AFPN [61] mitigates feature degradation across scales by stabilizing hierarchical interactions, whereas [40] proposes Triple Cross-Criss FPN (TCFPN), introducing a bidirectional residual mechanism for richer multi-scale integration and a feature aggregation module that compresses and redistributes spatial features.

D.Detection-Oriented Modifications for Small and Tiny Objects

In [50], deformable convolution replaces standard operators, improving regression accuracy on small aircraft. MSGD-YOLO [35] addresses information loss due to repeated downsampling by adding a small-target layer to GFPN and using MSGConv for lightweight multi-scale extraction. CSPStage modifications replace 3 × 3 convolutions with RepConv, which uses multi-branch training and single-branch inference via reparameterization for faster deployment. Triple Attention is added at the end of each fusion branch to refine spatial and channel features for small-target detection.

WILD-YOLO [6] partially replaces YOLOv7’s heavy ELAN module with FasterNet for efficiency, while [90] employs Bi-PAN-FPN for improved multi-scale merging throughout the detection pipeline.

E.Specialized Lightweight Integration Modules

SOD-YOLO [46] introduces the Lightweight Feature Integration Module (LFI), forming a dual-branch structure that merges features while minimizing element-wise operations. The combined DSDM-LFIM architecture enhances deep–shallow fusion by retaining distinct feature streams, while a P2 detection layer explicitly targets small objects. RTD-Net’s Feature Fusion Module (FFM) [115] simplifies YOLOv5’s PANet using a BiFPN variant with jump connections and weighted fusion to balance contributions across feature resolutions.

Finally, ASFF [5] enhances YOLOv3 by generating an adaptive weighted sum of resized feature maps, improving multi-scale feature blending, while E-FPN [102] expands the pyramid to four levels using Enhance Blocks for robust cross-scale detection. While NAS-based methods offer automated architecture generation, many still prioritize accuracy over efficiency, limiting their suitability for onboard deployment. Across all enhancements, the overarching challenge remains balancing computational efficiency with the need for robust small-object detection in complex UAV imagery [208]. However, new NAS methods have emerged for hardware-aware design.

#### 5.1.3. Head Optimization Strategies for Small Object Detection

For small object detection, several modifications have been made to the head. For instance, ref. [78] introduces a high-resolution detection branch for small objects, while in [79], YOLOv5 replaces the large object prediction head with one designed for tiny objects. In [40], feature mapping layers for large objects are pruned, and a shallow detection head with self-attention is added for small objects. Similarly, ref. [12] introduces an additional detection layer for small objects in YOLOv5s, while ref. [39] in YOLOv5-n replaces the original head with a new prediction head optimized for small UAV objects.

Additionally, ref. [91] in YOLOv5 introduces a new detection head with a stride of 4 for small vehicles. Furthermore, advanced detection heads like the RT-DETR head and IoU-aware query selection [200] have replaced traditional approaches, eliminating the need for Non-Maximum Suppression (NMS) and refining predictions for better small-object detection. Several studies have modified the original YOLOv8 and YOLOv5 loss functions to enhance object localization and detection accuracy, especially for small and occluded objects. The Normalized Wasserstein Distance (NWD) added with Generalized Intersection over Union (GIoU) to improve optimization weight and regression accuracy [12,89]. The Alpha-Complete Intersection over Union (α-CIoU) addresses the imbalance between positive and negative samples in aerial image detection [33,38], while the Focal-EIOU loss enhances localization precision [39].

The Robust Intersection over Union (RIOU) loss prevents calculation failures and ensures better shape consistency between predicted and ground truth boxes [6,36,40,81]. Additionally, the Single Stage Headless (SSH) context structure improves feature extraction, aiding in the detection of small or occluded objects [110]. For embedded processing, several studies have focused on optimizing the detection head to improve both inference speed and detection accuracy.

In RTD-Net [115], for instance, introduces the Attention Prediction Head (APH) to enhance the model’s capability to focus on relevant objects. To achieve this, the Normalization-based Attention Module (NAM) is designed as a lightweight attention mechanism that is integrated before the detection heads, forming what are known as Attention-Preceding Heads (APHs). NAM draws inspiration from CBAM but incorporates a redesigned structure for channel and spatial attention submodules. Specifically, NAM generates attention maps separately for both channel and spatial dimensions, which are then applied to the input feature map via element-wise multiplication. This selective emphasis enables the model to concentrate more effectively on salient regions, thereby improving object detection performance. In parallel, efforts have also been made to reduce the computational complexity of detection heads. The Efficient Decoupled Head aims to minimize inference costs by decreasing the number of intermediate convolutional layers, all while preserving the spatial dimensions of the input feature maps. This structure builds on the concept of the decoupled head architecture, initially introduced in FCOS [148] and later adopted by anchor-free models such as YOLOX [188]. The decoupling of classification and regression branches facilitates better network convergence and enhances prediction precision. However, this improvement typically comes with increased inference overhead due to the parallel branch design. To address this issue, further refinements have led to the development of a more lightweight decoupled head [21], characterized by fewer convolutional layers and reduced channel widths, to be able to achieve real-time performance. Additionally, to maintain strong regression performance without compromising speed, implicit representation layers have been integrated. Re-parameterization techniques are also employed in this architecture to streamline the model during inference, reducing the computational burden while maintaining detection accuracy.

#### 5.1.4. Loss Function

To improve detection accuracy, especially for small and challenging targets, a range of advanced loss functions have been proposed. In [50], EIoU replaces the traditional CIoU to enhance regression accuracy for small objects. Rep-ShuffleNet [48] incorporates BIoU Loss to improve localization precision, while refs. [35,97] adopts WiseIoU, which employs a dynamic, non-monotonic focusing mechanism to better balance loss contributions. SCAFPN [51] introduces Hybrid-Random Loss (HRL), specifically targeting the challenges of small object detection, and RTD-Net [115] applies L1 regularization to further refine localization accuracy.

Additionally, in [21], the training process is structured into three distinct stages to gradually enhance model performance. In the initial stage, conventional loss functions such as Generalized IoU (gIoU) for bounding box regression and Balanced Cross-Entropy for classification and objectness scoring are employed. In the second stage, HRL is introduced in combination with data augmentation approaches to improve the model’s robustness, particularly for small object detection. Finally, the third stage disables data augmentation to stabilize training, replaces gIoU with Complete IoU (CIoU) for improved localization precision, and applies L1 regularization to fine-tune the model. This progressive, multi-stage training strategy effectively balances accuracy, efficiency, and generalization across diverse detection scenarios.

#### 5.1.5. Lightweight Transformers for Real-Time Aerial Detection

Transformer-based detectors are increasingly adapted for aerial object detection due to their strong ability to model long-range dependencies and global spatial context in UAV imagery. However, standard DETR and ViT architectures remain computationally heavy because the self-attention mechanism scales quadratically with sequence length. This creates substantial memory pressure and latency, making real-time deployment challenging both onboard and even on desktop GPUs [171,209]. The core issue stems from the need for self-attention to compute relationships between all token pairs, resulting in significant computational and memory overhead as the spatial resolution increases [210]. These limitations have motivated a surge in research into lightweight, scale-aware, and frequency-preserving transformer designs suitable for aerial platforms.

Despite these constraints, several enhanced transformer-based detectors have demonstrated real-time or near-real-time performance in UAV scenarios. For instance, UAV-DETR [211] employs an RT-DETR backbone and integrates a Multi-Scale Feature Fusion and Frequency Enhancement (MSFF-FE) module to retain high-frequency texture, a frequency-focused downsampling (FD) module to preserve spatial structure during downsampling, and a Semantic Alignment and Calibration (SAC) module to improve feature consistency across different scales. These adaptations enhance the detection of small and occluded objects while maintaining real-time speed. Likewise, ref. [117] further extends RT-DETR by incorporating partial convolutions in the backbone and introducing three specialized modules tailored for UAV open-water detection: the Small-Object Enhancement Module (SOEM), the Cross-Scale Feature Pyramid Interaction Module (CFPIM), and Multiscale Sensing Fusion (MSSF). SOEM strengthens fine-grained local perception, CFPIM improves hierarchical multi-scale fusion, and MSSF models correlations between global and local contexts to better represent small and sparse targets.

Parallel efforts investigate integrating lightweight transformer components into established detection pipelines to capture long-range dependencies while reducing computation. In [51,115], the authors propose SCAFPN to mitigate inter-layer interference during feature fusion and introduce CSL-MHSA to obtain long-range contextual cues for small objects. They further reduce overhead through depthwise separable convolutions (DSConv), SPD conv, and improved SimOTA label assignment, with FP16 quantization using NVIDIA TensorRT to accelerate inference. Their fusion strategy employs a progressive Mild module that merges adjacent pyramid levels (P2–P5), avoiding dense connections and reducing semantic gaps and computational burden.

The design in [115] expands on this direction through the Enhanced Contextual Transformer Block (ECTB), which incorporates Convolutional Multi-Head Self-Attention (CMHSA) inspired by Swin Transformer to strengthen global reasoning in challenging scenes with clutter or occlusion. Recognizing the prohibitive cost of traditional self-attention, the authors introduce an additive attention mechanism that replaces quadratic key–value interactions with linear element-wise operations, significantly reducing the computational load while maintaining accuracy. This lightweight formulation enables attention to be used across all network stages. Building on these ideas, they develop the SwiftFormer family, which achieves favorable accuracy–speed trade-offs and is reported to run approximately twice as fast as MobileViT-v2 [212]. These approaches collectively illustrate an emerging trend: replacing or augmenting classical self-attention with more efficient approximations to enable deployment in computation-constrained UAV systems.

In addition to these enhanced transformer modules, lightweight DETR-style detectors also achieve promising performance on aerial tasks. Another UAV-DETR [213] integrates Channel-Aware Sensing (CAS), a Scale-Optimized Enhancement Pyramid (SOEP), and a Context–Spatial Alignment Module (CSAM), achieving 51.6% mAP@0.5 on VisDrone2019 at 30 FPS with only 16.8M parameters. AUHF-DETR [214] reaches 68 FPS on AGX Xavier in UAV simulation. AUHF-DETR is a compact real-time detector built on the RT-DETR framework and tailored for UAV-based remote sensing, where targets are often very small and hardware resources are limited. The model uses a lightweight WTC-AdaResNet backbone to extract multi-scale features efficiently, and replaces standard global self-attention with a more economical PSA attention design to keep computation manageable on embedded GPUs. A specialized BDFPN module is introduced to strengthen small-object representation and ease the difficulties that arise in one-to-one matching during training. In addition, a loss term that adaptively emphasizes small targets further improves localization accuracy. With around ten million parameters and high inference speed on devices such as the AGX Xavier, AUHF-DETR improves its superiority over YOLOv8m and YOLOv11m in accuracy and inference. The comparison shows that AUHF-DETR delivers the most efficient Transformer-based performance on the Jetson AGX Xavier. AUHF-DETR-S achieves 68 FPS, far exceeding RT-DETR-r18 (39 FPS), while using half the parameters and less than half the GFLOPs. The medium variant (AUHF-DETR-M) also surpasses RT-DETR-r18 with higher speed and lower computation. Although YOLOv8-M and YOLOv11-M remain strong CNN baselines, AUHF-DETR-S provides competitive real-time speed with substantially lower complexity, demonstrating that the proposed design enables Transformers to operate efficiently on embedded UAV hardware, making it suitable for onboard real-time UAV applications. VMC-DETR [215] also improves small-object detection by integrating a frequency-domain VHeat C2f module, a large-kernel MFADM for multi-scale aggregation, and a CAGFM spatial–context fusion strategy, achieving 9.2 ms per image. Similarly, SF-DETR [216] introduces a lightweight dual-scale transformer backbone (ScaleFormerNet) with frequency-fused enhancement networks, reaching 51.0% mAP@0.5 while outperforming RT-DETR (R18) in the authors’ experiments.

Together, these models illustrate a consistent set of architectural principles observed in the recent literature: (1) hybrid CNN–ViT backbones that preserve edge details while enabling global reasoning; (2) explicit preservation of high-frequency information through wavelets, heat conduction, or frequency fusion; (3) simplified decoders and efficient attention approximations; and (4) progressive, rather than dense, multi-scale fusion to reduce semantic gaps and computation. While the original papers do not explicitly conclude with a unified design philosophy, the trends inferred across their architectures and results suggest that lightweight transformer-based approaches are becoming increasingly viable for real-time UAV detection when carefully engineered for efficiency, detail preservation, and hardware constraints.

### 5.2. Pruning

Pruning is an effective technique in convolutional neural networks (CNNs) that removes redundant parameters while maintaining overall model performance. It is typically categorized into two types: unstructured pruning and structured pruning.

#### 5.2.1. Unstructured Pruning

Unstructured pruning operates at the individual weight level, removing less important weights by setting them to zero [58].

#### 5.2.2. Structured Pruning

On the other hand, targets larger components of the network, such as entire filters [60] or channels [59,61]. By preserving the model’s original structure, it maintains compatibility with standard inference engines and leads to actual reductions in computation and inference time. This type of pruning evaluates the importance of filters or channels based on various criteria. For instance, one method ranks filters using their L1-norm values and prunes the least important ones [59]. Another approach evaluates the rank of output feature maps to identify and eliminate less critical filters [125]. Some techniques, such as adaptive batch normalization [60], rely on repeated cycles of pruning and fine-tuning to select optimal configurations, although this can be computationally expensive. To mitigate the overhead of multi-phase training, recent research has explored pruning techniques that require only a single training pass. One such method introduces Zero Invariant Groups (ZIGs), which deactivate unimportant channels and their associated weights, ensuring the output remains consistent with that of a smaller network [60]. It also employs Column Subset Selection (CSS) to detect redundancy among filters and applies a pruning-regrowth strategy that adjusts filter importance dynamically throughout training. DepGraph [58], used in [45]. introduces a group-level importance criterion, learning consistent sparsity across dependent parameters. By first Organizing parameters into sets, then modeling their interconnections through a dependency graph, and finally applying pruning at the group level. In [62], the feature extraction layers dedicated to detecting large objects in the YOLOv8 model are removed, leading to a substantial decrease in computational load. With some modifications to the structure, the accuracy is improved with fewer computational demands. The pruning process in [5] involves sparse training to determine the importance of channels, followed by the selective removal of channels based on importance scores. A local safety threshold is used to prevent excessive pruning, ensuring that the model retains its functionality. After pruning models are fine tuned to maintain its effectiveness.

### 5.3. Quantization

Model quantization involves transforming a neural network’s floating-point computations into fixed-point representations, typically using low-precision formats. This process often entails converting standard FP32 (32-bit single-precision floating-point) to either FP16 (16-bit half-precision floating-point) or INT8 (8-bit fixed-point integer). These compression techniques aim to enhance efficiency by minimizing model size and reducing storage requirements.

#### 5.3.1. Post Training Quantization

TensorRT [51], deployed on GPUs, also helps accelerate performance through quantization and layer fusion [54,115,217]. TensorRT (NVIDIA Tensor Runtime) is a deep learning inference engine developed by NVIDIA that accelerates the execution of neural networks on NVIDIA GPUs. By optimizing models for inference, TensorRT delivers rapid performance, making it invaluable for applications requiring minimal latency and high throughput. Quantization strategies, including INT8 and FP16, have been widely explored to balance speed and accuracy. FP16 quantization has demonstrated significant improvements in inference speed, particularly on edge computing devices, while maintaining reasonable accuracy. However, despite its potential speed advantages, INT8 quantization can lead to substantial accuracy degradation in some cases [51,56]. FP32 quantization results are less favorable due to factors such as overhead from quantization/dequantization operations, unquantizable operators, and excessive memory usage on the shared hardware platform [51]. Additionally, techniques such as DSConv have been introduced to further optimize quantized models by utilizing low-cost integer operations while preserving kernel weights and output probability distributions. Replacing the standard convolution with DSConv optimized the model by quantizing the floating-point convolution kernel parameters and approximating their restoration, reducing computational resource requirements [45,55]. The impact of quantization on object detection models varies based on factors such as resolution. While reducing input resolution can enhance processing speed, it often comes at the cost of accuracy loss [127]. Methods like Selective Tile Processing (STP) have been shown to mitigate these drawbacks by maintaining localization quality while benefiting from quantization speed-ups. Automatic Mixed Precision (AMP) training, which combines FP32 and FP16 computations, has also been explored to reduce GPU memory usage and accelerate training with minimal accuracy trade-offs. Overall, FP16 quantization has proven to be a practical compromise between speed and accuracy, making it a practical choice for deployment.

In the context of aerial object detection, quantization is increasingly being explored to achieve real-time performance on resource-constrained hardware. For example, several recent UAV detection studies incorporate INT8 deployment pipelines or TensorRT-based FP16/INT8 engines to accelerate inference on Jetson-class processors [51,115,211].

#### 5.3.2. Quantization-Aware Training (QAT)

QAT is designed to mitigate the accuracy degradation that often occurs when neural networks are quantized after training. Instead of applying low-bit quantization only at inference time, QAT introduces fake quantization operations during the forward pass of training. These operations simulate the behavior of low-precision arithmetic (e.g., 8-bit) while keeping the underlying weights in floating-point format for gradient computation. By exposing the network to quantization effects during training, the optimization process adapts the weights to be more resilient to numerical approximation, reducing the accuracy drop commonly observed with post-training quantization.

During training, fake quantization tracks the activation ranges that each layer encounters, encouraging the network to constrain its internal dynamic range. This results in more stable quantization boundaries and improves robustness once the model is converted to a fixed-point representation. Although representing weights and activations with limited bit-width inevitably introduces numerical approximation, deeper models and well-regularized architectures tend to tolerate this reduction with minimal accuracy loss [218]. However, this technique is used for natural scene datasets but has not been tested on aerial detectors yet.

These works indicate that quantization-aware optimization can make transformer-based aerial detectors more feasible for onboard execution, particularly when combined with lightweight modules or efficient attention mechanisms.

### 5.4. Knowledge Distillation

Knowledge distillation is a technique where a smaller, simpler model (the student) is trained to mimic the behavior of a larger, more complex model (the teacher). This approach helps improve the performance of the student model, especially on tasks with limited computational resources. In [63] knowledge distillation method is combined with a self-supervised distillation approach, with speed 311.5 FPS ON RTX 3090Ti. However, it is challenging for lightweight networks to learn effective visual features from large-scale complex datasets [64].

### 5.5. Neural Architecture Search (NAS) and Real Time

Neural Architecture Search (NAS) automates the design of neural networks and has become a central tool for producing architectures that satisfy both accuracy and efficiency. Traditional NAS approaches (RL or evolutionary controllers) are computationally expensive; more recent families focus on hardware-aware, efficient search and on one-shot, weight-sharing methods, which dramatically reduce the search cost while producing architectures that meet target latency and memory budgets [219].

A few practical NAS paradigms stand out for lightweight design:

#### 5.5.1. Differentiable (Hardware-Aware) NAS

Differentiable NAS (FBNet style) [220] formulates architecture selection as a continuous optimization and incorporates measured operator latency into the search objective, enabling direct hardware-aware optimization for mobile targets. FBNet demonstrated that differentiable search with latency lookup tables can find ConvNet blocks that outperform hand-designed mobile nets on measured latency for target devices

#### 5.5.2. Direct Hardware-Aware Search

ProxylessNAS [221], removes proxy datasets and allows searching directly on the target task and hardware, using path binarization and latency-aware regularization to produce specialized mobile networks with competitive accuracy and measured latency gains on mobile GPUs/CPUs.

#### 5.5.3. One-Shot/Weight-Sharing and Specialized Deployment Once-for-All (OFA)

The OFA [222] network is designed to enable efficient deployment of neural networks across a wide range of devices and resource constraints without the need to train a separate model for each scenario. Instead of designing or training individual networks for different hardware or latency requirements, which is computationally expensive and environmentally costly, OFA decouples the training and search processes. A single, pre-trained OFA network can support a vast number of architectural configurations, allowing a specialized sub-network to be quickly selected to meet specific device or latency constraints without additional training. This flexibility is achieved through a progressive shrinking algorithm, which generalizes pruning across multiple dimensions, depth, width, kernel size, and input resolution, producing an enormous number of sub-networks that retain high accuracy.

#### 5.5.4. Search Efficiency and Evaluation Methods

A separate but related strand of work focuses on efficient evaluation mechanisms (predictors, low-cost proxies, early stopping, and one-shot weight sharing) to make NAS feasible for real operational tasks. Surveys and methodological papers summarize how these techniques reduce GPU hours and make NAS practical for edge deployment [223].

The Inception-ResNet network has been improved with a Fast Architecture Search (FAS) module that automates the identification of optimal feature fusion paths and mitigates information loss during multi-layer extraction of dense, rotated targets. Anchor-free, decoupled detection heads have been proposed to separate bounding box regression from classification, enabling specialized feature learning and improved detection efficiency for densely packed or non-horizontal objects. Complementary techniques such as ellipse center sampling (ECS) expand the sampling area for rotated targets and convert rotated bounding boxes into a distance-based representation using an elliptical Gaussian distribution, facilitating gradient computation for arbitrary IoU angles. These approaches rely primarily on convolutional layers, reducing model complexity and enhancing deployability on edge platforms. Experimental evaluations on datasets such as DOTA and HRSC2016 demonstrate that these methods achieve improved accuracy and efficiency, highlighting their suitability for the unique challenges of aerial object detection [224].

YOLO-NAS has been specifically developed to enhance small-object detection, improve localization accuracy, and optimize the performance-to-compute ratio, making it particularly well-suited for real-time applications on edge devices. Its open-source design further supports research and experimentation. A key innovation of YOLO-NAS lies in its integration of quantization-aware modules, known as QSP and QCI, which employ re-parameterization techniques to enable 8-bit quantization while minimizing accuracy loss during post-training quantization. Additionally, the model incorporates a hybrid quantization strategy that selectively applies quantization to certain components, striking a balance between latency and accuracy rather than uniformly affecting all layers [225].

Neural Architecture Search (NAS) is regarded as essential for the development of efficient and robust perception models used in UAV and drone platforms, where real-time small-object detection and strict computational limitations remain major challenges.

#### 5.5.5. Other Real-Time Research

In [120], A three-step approach has been developed and deployed within a Docker runtime environment on an Nvidia Jetson AGX Xavier board. To enable rapid object detection, the captured images are divided into K segments, each processed in parallel within separate containers running the YOLOv5 object detection algorithm. The final detection results are assembled from the individual K detections. Experimental findings strongly support the method’s efficiency, demonstrating near-real-time object detection for ultra-high-resolution (8K) images, with processing times of less than 1 s per frame.

In [57], The E3-UAV system includes three main components: the user space, decision module, and model running library. The user space allows configuration of UAV performance, model settings, and task metrics. The decision module optimizes flight parameters by balancing energy consumption with task needs, adjusting factors like altitude, speed, and detection model. It calculates energy use and selects the optimal configuration given trade-offs. The model running library supports YOLOv3 deployment on TensorRT, TensorFlow, and PyTorch for object detection. E3-UAV integrates with UAV control systems for real-time data processing and task management, focusing on energy efficiency and performance optimization.

In [118], the UAV-ODS system has three main modules: data acquisition, data processing, and data presentation. The acquisition module collects video, GPS, altitude, and speed data, which is sent to a ground control system for target detection. The processing module uses an RTMP server for real-time streaming and applies a YOLOv5s detection model to identify targets, with results sent to a server for display. The front-end module presents real-time detection and UAV flight positions. The system uses a dataset of 603 images from 16 locations in Wuhan for outfall detection, employing YOLOv5s for its speed, efficiency, and accuracy. The model’s backbone includes Focus and CSP modules, with FPN and PAN structures for multi-scale feature fusion. Post-processing, including thresholding and NMS, refines detection results.

**Table 10 sensors-25-07563-t010:** Summary of recent Object Detection Model architectures, tested on edge and onboard platforms.

Ref.	Base Model	Backbone	Neck	Head	Loss	Opt.	Platform
[46]	yolov7	The DSDM-LFIM backbone enhances small object detection by combining efficient deep-shallow feature extraction (DSD) with lightweight dual-branch feature fusion (LFI)	Original multi-scale feature fusion	Adds a high-resolution P2 branch to improve small-object detection. Uses K-means to optimize anchors per detection head for the VisDrone dataset	Same as the Base model	Scaling the number of channels by 0.2	1.4 M parameters, mAP50 33.4%, 36.6 FPS (VisDrone) inference on edge Devices (Atlas 200I DK A2)
[5]	YOLOv3	IRFM expands the receptive field using multi-scale dilated convolutions and fuses outputs via learnable weights and shortcut connections for better small object detection	Adaptively Spatial Feature Fusion (ASFF) to enhance multi-scale representation	Anchor optimization integrated into training loop using dynamic anchor generation to maximize IoU	N/A	Deconvolution replaces nearest-neighbor upsampling. Coordinate decoding handled outside the model. Quantized to 8-bit using the uds710 tool. NMS and decoding are implemented in C++ for execution on the NPU.	mAP50 89.7%, 35.7 FPS inference on T710 NPU (Neural Processing Unit) (UAV car custom datast)
[115]	YOLOv5	LFM: Uniform multi-branch design to reduce computation. ECTB: CMHSA for global feature capture and occlusion handling	Eliminates redundant nodes (e.g., P1tm, P4tm) to simplify the structure. Adds shortcut connections for better feature propagation. Introduces learnable weighted fusion to adaptively emphasize informative features.	The attention prediction head (APH) is designed based on the NAM attention mechanism to improve the ability of the model to extract attention regions in complex scenarios	As the base model	N/A	21.7 M parameters, 33.4 inference FBS on NVIDIA Jeston TX2 (UAV air custom dataset)
[50]	YOLOv5n	– Combines ShuffleNet v2 with YOLOv5n.– Introduces a Coordinate Attention (CA) module at the end of the backbone to enhance spatial and orientation – Includes a custom CBRM module—composed of Conv, BatchNorm, ReLU, and MaxPool layers—for efficient feature extraction.	Same as Base model	Same as Base model	Replaces CIoU with EIoU to improve bounding box regression and accelerate convergence, especially for small-scale aircraft targets.	N/A	0.9 M parameters, mAP50 84.8%, 22.6 FPS (post + preprocessing + inference) NVIDIA Jetson Xavier NX (MAR 20 dataset)
[6]	YOLOv7	– Replaces certain ELAN modules in YOLOv7 with the lightweight G-FasterNet, combining FasterNet and GhostNet to reduce parameters and memory usage. – GhostConv is used in place of standard Conv to preserve feature extraction efficiency with lower computational cost.	– Replaces certain ELAN modules in YOLOv7 with the lightweight G-FasterNet, combining FasterNet and GhostNet to reduce parameters and memory usage.– GhostConv is used in place of standard Conv to preserve feature extraction efficiency with lower computational cost.	SimAM attention in the head, faster net use partial convolution (PConv) scenarios	Same as Base model	N/A	13.38 M parameters mAP50 95.04% on Jetson Nano, Jetson Xavier NX NC S2 (WAID dataset)
[102]	YOLOv4	Same as Base model	– E-FPN constructs a 4-level pyramid (F2–F5) for enhanced multiscale feature exchange. – An Enhance Block at the input improves semantic representation by splitting features into low- and high-resolution branches with depthwise conv and CBAM – A Refine Attention module at the output mitigates aliasing effects from repeated fusion, improving detection of small objects across scales.	– PixED Head uses a spatial-channel encoder–decoder with pixel-encode (PE) and pixel-decode (PD) to boost tiny object detection efficiency. – A Feature Extraction Module (FEM) with depthwise conv, pointwise conv, and CBAM refines features. – An auxiliary head (Aux Head) aids sample assignment during training only, adding no inference cost.	– Improved SimOTA label assigner with CIoU and Focal Loss addresses class and aspect-ratio imbalance. PLA loss aligns features between heads.	N/A	0.7 M parameters, mAP50 22.7%, 103 FPS on NVIDIA Jetson Xavier NX, and 24.3 FPS on Jetson Nano GPU
[49]	YOLOv4	MobileNetV3 is used to replace the original feature extraction network CSPDarkNet53 network	– SPP+PAN+YOLO Head structure of YOLOv4 is still used in neck and head. – A portion of the original 3 × 3 standard convolutions in PANet is replaced with depthwise separable convolutions. This substitution reduces both computational cost and the number of parameters to approximately one-fourth of those in YOLOv4.	MobileNetV3 and self-attention are integrated to enhance feature extraction, while Softer-NMS replaces DIoU-NMS to address the mismatch between classification confidence and localization accuracy. Rather than discarding overlapping boxes, Softer-NMS reduces their confidence scores and predicts localization confidence, resulting in more accurate detections	Softer-NMS replaces DIoU-NMS	N/A	23.8 FPS Nvidia Jetson TX2, 9.6 FPS on Raspberry Pi 4B
[48]	YOLOv8	Use Rep-ShuffleNet, based on ShuffleNetv2 to improve the original backbone of YOLOv8s, add the lightweight channel attention mechanism ECANet,	Same as Base model	Same as Base model	Binary Cross-Entropy (BCE) Loss is used for classification, while box regression combines Distribution Focal Loss (DFL) and CIoU Loss. – To enhance accuracy, CIoU is improved to BIoU Loss, which directly compares the actual aspect ratios of predicted and ground truth boxes, rather than relying on a relative aspect ratio similarity. This shift from approximate to precise comparison improves the model’s localization accuracy.	Building on edge intelligence and federated learning concepts, the FI framework and the multilayer collaborative federated learning (MLC-FL) algorithm for efficient federated learning are introduced. – By using asynchronous communication and low-frequency data exchange, MLC-FL enables local models to optimize automatically and efficiently. This upgrades the traditional coal mine IoVT system into an intelligent, self-learning system.	7.8M parameters, mAP50 94.6%, 21.6 FPS on NVIDIA Jeston AGX Xavier (CMUOD, survillance)
[92]	YOLOv8	Deformable Separable Convolution Block (DSCBlock), separating feature channels, a channel weighting module is proposed. This module calculates weights for the separated feature map, facilitating information exchange across channels and resolutions. Moreover, it compensates for the effect of point-wise (1 × 1) convolutions. A 3D channel weighting module is introduced to efficiently extract features by applying weighting operations along the channel dimension, avoiding the high cost of 1 × 1 convolutions, compensating for the accuracy loss with the efficient feature modeling capability of DCNv2	The PA-FPN-CSPD framework introduces adaptive sampling and a novel channel weighting module to enhance feature interaction. Instead of using costly pointwise (1 × 1) convolutions, the channel weighting module operates along the third (channel) dimension, enabling efficient filtering of key features while reducing the impact of deformations. To strengthen information exchange across layers, it calculates adaptive weights for separated feature maps. Additionally, the newly designed lightweight network structure, named Cross-Stage Partially Deformable Network (CSPDBlock), built around the DSCBlock, further establishes multidimensional feature correlations, improving the representation and robustness of each layer.	Same as Base model	Same as Base model	N/A	8.6 Mparameter, mAP50 34.2%, 24.7 FPS, Jetson Xavier NX (VisDrone dataset)
[51]	YOLOv5n	optimization of the entire network using DSConv. Additionally, integrating SPD Conv, which is sensitive to small targets, using Cross-Space Learning Multi-Head Self-Attention mechanism, enhancing the C3 module by using (Distribution Shifting Convolution). DSConv achieves lower memory consumption and higher computational speed.	Sparsely Connected Asymptotic Feature Pyramid Network (SCAFPN), introduces a sparse, asymptotic fusion strategy. It starts by merging adjacent low-level features and gradually incorporates higher-level features using an intermediate “Mild” module, which performs upsampling, weighted fusion, and 1 × 1 convolutions. This design limits fusion to neighboring layers, reduces parameter redundancy, preserves semantic integrity	label assignment strategy using SimOTA	Use the SimOTA label assignment strategy	employed the Tensor Runtime engine (NVIDIA TensorRT) to perform FP16 quantization	5.18 M parameter, mAP5034.8%, 35 FPS on NVIDIA Jetson Xavier Nx edge (VisDrone dataset)
[11]	YOLOv7	Same as Base model	Same as Base model	using a decoupled regression detection head	Combining Generalized IoU loss for precise localization, and balanced cross-entropy losses for objectness and classification to handle class imbalance. It also introduces a Hybrid Random Loss strategy during training to improve the detection of small objects	This paper combines both lossy reduction and lossless reduction (re-parametrization). To enhance small object detection, a scaling and stitching approach is proposed in data augmentation and redesigns the loss function to focus more on small objects, FP16-precision with TensorRT	40.5 M parameter, mAP50 44.8%, FPS inference on Nvidia Jetson AGX Xavier (VisDrone dataset)
[121]	Mask R-CNN-ResNet18, YOLOv8, SSD-MobileNet	Same as base model	Same as base model	Same as base model	Pre-processing with Sobel–Feldman filter, Enhances contrast along object boundaries, emphasizes edges for better feature extraction, reduces background noise, and improves the visibility of small objects in complex scenes	N/A	83.3 FPS inference for YOLOv8, 47.6 FPS for mask R-CNN, 62.5 FPS inference for SSD-MobileNet on NVIDIA Jetson Nano, 2.56 FPS inference for YOLOv8, 17.86 FPS for mask R-CNN, 2.87 FPS inference for SSD-MobileNet on Rspbirry PI 4B
[34]	YOLOv4 and YOLOv7	Same as Base model	Same as Base model	Same as Base model	Same as Base model	– A cloud–edge hybrid architecture in which AI tasks are handled locally at the edge, while the cloud is used for data storage, processing, and visualization. – TensorRT accelerator	38–40 FPS on Jetson Xavier AGX edge (2688 × 1512 resolution) and 8–10 FPS for (3840 × 2160) resolution custom dataset
[100]	YOLOv7-Tiny	Same as Base model	Same as Base model	truncated NMS (Non-Maximum Suppression) is used	Manhattan Intersection over Union (MIOU) loss	Satellite images are first divided into smaller tiles, and cloud-covered regions are filtered using the PID-Net method. The remaining clear tiles are then processed using a YOLOv7-Tiny model enhanced with MIOU loss to detect remote sensing objects. Finally, the results are mapped back to their original positions in the full image. Use TensorRT	mAP50 76.9%, TensorRT-FP16 160 FPS on NVIDIA Jetson AGX Orin. Cloud (latency 8.3 ms object detection 6.3 ms, Post Processing 31.6 ms, Total 21.6 FPS (DOTA dataset)
[119]	SSD	mobilenet replaces VGG-16 or ResNet	Same as Base model	Same as Base model	Same as Base model	N/A	mAP50 92.7%, 26 FPS on NVIDIA Jetson Nano, 18 FPS on Raspberry Pi 3 B Fire detection
[93]	YOLOv5, YOLOv6	EfficientRep, (RepBlock, RepConv)	Rep-PAN (RepBlock, RepConv)	Same as Base model	Same as Base model	N/A	N/A
[125]	YOLOv8	MaxPooling+Ghost Convolution	PAFPN, CoordBlock includes coordinate attention and CoordConv to enhance features and reduce the loss of spatial information. Additionally, Partial Convolution (PConv) directly before the detection head.	Extra head for small objects	N/A	nano variant achives 1.2M parameters, mAP50 39.7%, 56 FPS with FP16 b = 1, 147FPS FP16 B = 16 inference on Jetson AGX Xavier. (VisDrone)	
[57]	YOLOv3	Same as base model	Same as base model	The anchor boxes are resized proportionally to match different input resolutions, ensuring optimal performance for each model configuration. For example, an anchor box of (2, 5) at a 416 × 416 resolution would be adjusted to (4, 10) for an 832 × 832 resolution.	Same as base model	TensorRT inference engine with 16-bit quantization.	less than 0.1 ms latency on NVIDIA Jetson Xavier NX. (VisDrone)
[56]	YOLOv3	Same as base model	Same as base model	Same as base model	Same as base model	– Joint Quantization – Reduces the precision of weights and activations to lower bit-widths, thereby minimizing memory and computational requirements.– Tiling – Splits high-resolution images into smaller tiles to enable processing on limited hardware without sacrificing detection accuracy. Quantization	quantization speed-up baseline by 1.35 on Jetson TX2. NVIDIA Jetson TX2 (352 × 352 input)
[120]	YOLOv4	Shallower CPSDarkNet53, parameters are shared between the object detection and semantic segmentation tasks	Same as base model	adding a segmentation head to an object detector backbone	Same as base model	N/A	Nvidia Xavier NX
[35]	YOLOv8	– Integrating the Ghost module and dynamic convolution into the CSP Bottleneck with two convolutions (C2f). – Spatial Pyramid Pooling with Enhanced Local Attention Network (SPPELAN) replaces Spatial Pyramid Pooling Fast (SPPF) to expand the receptive field	Multi-Scale Ghost Convolution (MSGConv) and Multi-Scale Generalized Feature Pyramid Network (MSGPFN). – Triple Attention39 is applied at the end of each information transfer branch to enhance the extraction of small-target information before sending the features to the network head	DyHead enhances detection precision for small targets. By incorporating three self-attention mechanisms into the detection head, DyHead redefines the four-dimensional tensor L×H×W×C as a three-dimensional tensor L×S×C. This approach applies scale-aware, space-aware, and task-aware attention in the L, S, and C dimensions, respectively.	Same as base model	N/A	2.6 M parameters, mAP50 45.2%, 24.6 FPS on Nvidia Jetson Orin Nano
[118]	YOLOv5	The base model structure. use input Pixel-level and spatial-level augmentations, along with object mosaic and background fusion.	Same as base model	Same as base model	Same as base model	Use TensorRT	24-33 FPS inference on NVIDIA Jetson AGX Xavier (Outfall)
[214]	RT-DETR	Partition Split Spatial Attention (PSA) replaces global self-attention → local ROI attention with high/low-frequency decomposition.	Bidirectional Dynamic Feature Fusion Pyramid Network (BDFPN) adds multi-scale bidirectional fusion with learnable dynamic weights, supplies denser supervision for small targets	Same as base model	remains RT-DETR-like but equipped with Inner-MPDIoU instead of L1 + GIoU (with a small-object penalty). The MPDIoU loss combines three components: an IoU-based overlap term, a minimum point–distance metric, and a normalization factor.	same as base model	10.29 M parameters, mAP50 53%, 68 FPS inference on NVIDIA Jetson AGX Xavier (VisDrone)

These studies collectively demonstrate that lightweight backbones, efficient feature fusion techniques, optimized detection heads, refined loss functions, and attention mechanisms contribute to improved object detection models, balancing accuracy and computational efficiency.

The models summarized in Table 10 collectively illustrate how recent research balances the competing demands of accuracy, computational cost, and real-time feasibility on resource-limited UAV platforms. A clear trend is the continual reduction in model parameters while maintaining acceptable detection performance. Many recent works adopt lightweight backbones such as ShuffleNetv2, GhostNet, Rep-ShuffleNet, or MobileNet, and replace standard convolutions with depthwise separable variants or Ghost modules. These strategies reduce FLOPs and memory consumption, yielding compact models (often under 2M parameters) that can achieve 20–100 FPS on devices such as Jetson Nano, Xavier NX, and AGX Xavier. However, this compression inevitably limits representational capacity, yielding lower mAP values than heavier detectors such as YOLOv7, YOLOv8-s, or Mask R-CNN, which achieve superior accuracy but require more powerful embedded hardware to sustain real-time inference. Another notable trade-off across the table concerns input image resolution. Higher resolutions enhance the visibility of small and dense objects, improving detection recall, yet they significantly increase computational burden and reduce throughput. Several studies mitigate this by adjusting anchor scales, adopting tiling strategies, or adding high-resolution branches such as P2 layers to preserve fine details without increasing input dimensions.

Because small-object detection remains one of the most significant challenges in UAV imagery, many architectures emphasize improved multi-scale representation and attention mechanisms. Enhanced feature pyramids (including E-FPN, SCAFPN, and MSGPFN), deformable convolution operators, pixel-encoder heads, weighted fusion layers, and lightweight attention modules such as NAM, SimAM, ECANet, or Triple Attention are frequently incorporated. These additions enhance the discrimination of small targets and robustness under occlusion or low contrast, though they often increase computational cost in the neck and head, leading to moderate FPS reductions when deployed on edge devices.

The influence of hardware constraints is also strongly reflected in the reported inference speeds. Devices such as Jetson Nano typically operate below 40 FPS unless the backbone is extremely lightweight, whereas Xavier NX and AGX Xavier offer sufficient capacity for mid-sized YOLOv5/YOLOv7 variants, achieving 20–103 FPS depending on the specific architecture and input resolution. The newer Orin series supports higher throughput, allowing the deployment of more complex heads and multiscale modules. Conversely, Raspberry Pi and low-power NPUs impose strict limits, making MobileNet-based SSD or highly compressed detectors the only viable options for real-time performance. These results demonstrate that model selection in UAV detection is inherently hardware-dependent.

Deployment-aware optimizations constitute another consistent theme. Many studies apply FP16 or INT8 quantization, TensorRT compilation, operator fusion, or reparameterization techniques to achieve substantial speed improvements, often doubling or tripling FPS with minimal or no loss of accuracy. Some works even redesign the entire decoding pipeline or integrate low-level C++ operations on NPUs, highlighting a shift toward co-optimizing the neural architecture and the deployment framework. As a consequence, the final achievable performance is increasingly determined not only by the network design but also by the compatibility between the model’s computational structure and the underlying hardware acceleration tools.

Based on the comparative analysis reported in [34], the inference-speed results show a consistent relationship between image resolution, model complexity, and hardware capability. On the RTX 8000, YOLOv7 in PyTorch maintains the highest throughput, reaching peak values of roughly 45–50 FPS at 1920 × 1080, demonstrating that desktop-class GPUs can sustain near–real-time performance even with heavier models. In contrast, Jetson platforms exhibit a sharp decline in speed as resolution increases. For example, YOLOv4-TRT on the Xavier AGX achieves noticeably higher FPS at 2688 × 1512 compared with 3840 × 2160, reflecting the increased computational load of processing more than twice the number of pixels. The Xavier NX shows the lowest speeds across all settings, with YOLOv7-PyTorch and YOLOv7-TRT generally operating below 10 FPS. The effect of cloud communication, sending detection results to the cloud, significantly reduces throughput, with the average speed dropping from about 12.3 FPS locally to around 5 FPS when cloud transmission is enabled. This demonstrates how network overhead can become a major bottleneck and highlights the need for efficient communication strategies in edge–cloud systems.

### 5.6. Leveraging CLIP Embeddings for Aerial Real-Time Visual Recognition

Contrastive Language–Image Pre-training (CLIP) is a vision-language model that learns a shared embedding space for images and natural language. The model employs two separate encoders, one for images and one for text, which are trained to project corresponding image–text pairs into a shared embedding space. During training, the similarity between matched pairs is maximized while that of mismatched pairs is minimized using a symmetric cross-entropy objective over the batch-wise similarity matrix [226]. This objective, closely related to the InfoNCE loss formulation [227], encourages semantic alignment between modalities. As a result, a pretrained CLIP model is capable of zero-shot classification, where candidate class labels are converted into text prompts, encoded by the text encoder, and the label whose embedding exhibits the highest cosine similarity with the image embedding is selected. This approach has been demonstrated to generalize effectively across diverse datasets without requiring additional task-specific fine-tuning [226].

Since CLIP is trained using natural language supervision rather than a fixed set of categorical labels, the resulting image–text embeddings capture a broad semantic space, enabling generalization to previously unseen visual concepts without requiring additional labeled data [226]. This characteristic renders CLIP and related vision–language models particularly suitable for open-vocabulary tasks, in which detectors or classifiers must accommodate categories that are not present during the original supervised training. Recent surveys and reviews have emphasized that large-scale vision language pretraining has emerged as a dominant approach for achieving open-vocabulary and zero-shot performance across multiple vision tasks, including classification, object detection, and semantic segmentation [228,229].

Researchers have adapted the CLIP paradigm for remote sensing either through domain-specific pretraining or by using CLIP as a semantic teacher. For example, RemoteCLIP [230] converts heterogeneous annotations (bounding boxes, segmentation masks) into caption style supervision and scales remote-sensing image–text datasets, producing embeddings better aligned with aerial scene semantics and improving downstream detection and segmentation tasks.

Building on RemoteCLIP, CastDet, an open vocabulary aerial object detection framework, uses a student teacher self learning paradigm in which a frozen RemoteCLIP model acts as a teacher to guide a detector during training. This external teacher provides semantic embeddings that refine both class-agnostic proposals and pseudo-labels for novel object classes unseen during supervised training. To maintain high-quality pseudo-labels, CastDet includes a dynamic label queue, ensuring that only reliable labels (as judged by RemoteCLIP) persist in training, which helps stabilize learning [231].

To enable real-time performance, several optimization strategies can be applied that reduce model size and computational demands without significantly affecting accuracy.

#### CLIP Optimization Techniques and Related Studies

Knowledge Distillation and efficient backbone replacementOne approach is to train a smaller “student” model using knowledge from a larger pre-trained CLIP “teacher.” Techniques like those explored in CLIP-Knowledge Distillation (KD) CLIP-KD [232] show that guiding the student to replicate the teacher’s feature embeddings (feature-level mimicry) is particularly effective. This allows the student model to maintain strong cross-modal alignment for zero-shot tasks while requiring far fewer resources.CLIP’s original design uses large Vision Transformers or ResNets, which are computationally heavy. By distilling knowledge into a student model, it is possible to adopt a lighter backbone, such as MobileViT or compact convolutional networks [232]. This flexibility is critical for aerial deployment, as smaller backbones drastically reduce computational cost while preserving the model’s ability to align images and text.Quantization and PruningFurther efficiency can be achieved by reducing the precision of model weights (quantization) and removing unnecessary parameters (pruning). For instance, TernaryCLIP [233] is a computationally efficient framework that reduces the precision of both the vision and text encoder weights in CLIP by representing them in ternary format, rather than using standard full-precision or floating-point values. It compresses both vision and text encoders to ternary weights while applying distillation-aware training to maintain performance. Combining pruning with quantization forms a multi-stage compression pipeline, yielding a compact, fast, and energy-efficient model capable of real-time operation.

In [116], while the model is still large, adapting CLIP for lightweight aerial systems is a promising and active research direction. The combination of domain-specific pre-training (e.g., RemoteCLIP), open-vocabulary detection frameworks (e.g., CastDet), and model compression techniques (distillation, quantization, pruning) creates a credible pathway for deploying vision-language models on resource-constrained platforms such as UAVs. However, further empirical research is needed, particularly on integrating lightweight networks and modules that have proven efficient in object detection models.

YOLO-World is a framework that has recently emerged, extending the traditional YOLO object detection paradigm to handle open-vocabulary detection. Unlike standard YOLO models, which are limited to a fixed set of categories, YOLO-World leverages vision-language modeling and pre-trained embeddings (e.g., CLIP embeddings) to recognize objects described by arbitrary text labels. This approach allows the detector to generalize to unseen categories without additional training (zero-shot capability). YOLO-World maintains the efficiency and speed characteristic of YOLO models while integrating language-guided supervision [234], achieving real-time performance on NVIDIA V100 platform with TensorRT.

### 5.7. Integrating NAS Methods with LLM Models for Edge Devices

In this study, a framework named PhaseNAS is proposed for detection tasks. Within PhaseNAS, LLM reasoning capacity is dynamically adjusted across exploration and refinement phases, and a structured template-based prompt language is introduced so that natural-language instructions can be translated into executable model configurations. A zero-shot detection metric is also developed to allow candidate YOLO-based architectures to be rapidly screened without the need for full training [235]. The search process is conducted within a constrained design space and is guided by three key principles. First, all architectural components are selected from predefined functional groups, such as convolutional layers or residual blocks, to ensure modularity. Second, consecutive modules must maintain dimensional consistency through strict channel alignment, enabling seamless integration throughout the network. Finally, the computational complexity of the generated architectures is restricted to predefined limits, ensuring that the resulting designs remain practical for deployment across a variety of hardware platforms. Through experiments conducted on different datasets, including VisDrone2019, it is demonstrated that superior architectures are consistently identified by PhaseNAS, with search time reduced by up to 86% while gains in accuracy and computational efficiency are achieved. On UAV detection benchmarks, YOLOv8 variants generated by PhaseNAS are shown to obtain higher mAP with reduced resource cost. PhaseNAS supports resource-adaptive model design, enabling UAVs and edge devices to perform real-time.

## 6. Limitation, Open Challenges and Future Directions in Onboard Real-Time Aerial Object Detection

### 6.1. Limitations of the Current Research

Current real-time aerial detection studies still face several structural and methodological limitations that restrict their suitability for UAV onboard deployment.

First, existing surveys and many recent studies focus primarily on detector families and high-level accuracy comparisons, yet provide only limited analysis of the end-to-end onboard pipeline, including preprocessing, postprocessing, quantization strategies, TensorRT deployment, and cloud–edge latency. This makes it difficult to translate algorithmic progress into deployable UAV solutions.

Second, lightweight and hardware-aware design remains underexplored; most works mention backbone efficiency but rarely examine trained quantization, structured pruning, re-parameterization, or on-device memory constraints, despite their direct effect on FPS and energy usage on Jetson-class devices.

Third, although Transformers are increasingly used to address fine-grained discrimination and long-range context, they often introduce heavy computation, and only a few studies redesign attention modules for edge devices. Similarly, real-time methods rarely evaluate performance on high-resolution datasets or fine-grained benchmarks such as Mar-20, thereby limiting understanding of robustness to complex scene variation.

Fourth, lack of consistent hardware-aware evaluation. Most studies benchmark FPS on desktop GPUs rather than on embedded platforms such as Jetson Nano, Xavier, Orin, EdgeTPU, or NPUs, making it difficult to translate these results to onboard UAV performance. Without measurements of pre/post-processing latency, the real-time suitability of many algorithms remains uncertain. Similarly, there is no standardized protocol for tiny-object evaluation: Definitions vary across datasets. This inconsistency hinders fair comparison and slows progress in developing reliable small-object detection methods.

Finally, modern topics essential for aerial settings like UAV-oriented NAS, adapting CLIP embeddings, domain generalization, or hardware model trained optimization are largely absent from previous surveys and only partially addressed in current literature. These gaps highlight the need for more comprehensive, deployment-oriented research.

### 6.2. Open Challenges

Real-time aerial detection must address a complex interaction between data characteristics, algorithm design, and hardware constraints. Several open challenges continue to limit consistent, reliable onboard performance:Small-Object Detection and Fine-Grained Discrimination Small objects in aerial imagery often lose essential visual details due to high-altitude imaging, leading to blurred edges, weak textures, and low-resolution feature maps. As a result, detectors struggle to capture fine-grained cues and maintain high recall.Multiscale Variation and Object Orientation Aerial scenes exhibit extreme scale variation and arbitrary object orientations, in which tiny targets lose texture during downsampling and large objects dominate feature learning. At the same time, rotated objects frequently misalign with horizontal bounding boxes, reducing localization accuracy. These factors jointly complicate feature extraction, multiscale fusion, and stable regression, making real-time onboard detection particularly challenging for lightweight models.Loss Function Limitations for Tiny Objects Conventional detection losses (e.g., IoU-based and focal variants) tend to favor large objects, produce weak or unstable gradients for tiny targets, and often fail to assign reliable positives to small boxes. As a result, small objects are frequently overlooked or poorly localized.Transformer complexity and limited edge adaptation Transformers provide strong global context but are computationally expensive, particularly for multi-scale features and high-resolution inputs. Only a limited number of works have redesigned attention mechanisms or used hybrid CNN–Transformer backbones that are efficient enough for real-time UAV deployment.Lack of unified compression and adaptive modeling strategies Quantization, pruning, NAS, and CLIP-based distillation are often applied individually and post-hoc. However, unified compression-first pipelines are rarely investigated, such as quantization during training rather than post-training.High-resolution and dense-scene processingMultimodal datasets gap and onboard multimodal detection are still an open challenge.Hardware and energy constraints on UAV processors. UAV onboard processors, such as Jetson Nano/Orin, NPUs, and embedded GPUs, face strict power limits (5–15 W) and limited memory bandwidth and cache sizes. Achieving real-time detection at ≥30 FPS under these constraints requires hardware-aware design, which remains insufficiently explored in many publications.Limited aerial datasets for fine-grained, multimodal detection and general datasets like COCO for aerial detection.

### 6.3. Future Directions

The extensive range of object detection tasks in aerial imagery imposes considerable computational demands, significantly affecting the efficiency of detection models. Consequently, the development of lightweight network architectures has become a central focus in current research. The primary goal of these models is to reduce computational complexity and minimize the number of parameters without compromising detection accuracy. Several strategies have been proposed to achieve this objective:Lightweight convolutional backbones. Backbones will continue moving toward operators such as depthwise convolution, Ghost modules, PConv, channel-split and shuffle, SPDConv and DSPConv.Efficient multiscale and attention mechanisms. Developing more efficient attention modules (ECA, Coordinate Attention, NAM, SimAM) and lightweight multiscale fusion blocks will play a growing role in addressing dense and tiny objects. These modules preserve spatial–channel interactions with minimal computational overhead.Lightweight and transformer detectors. Recent advances in Transformer-based object detection show a clear shift toward architectures that blend CNNs with ViTs to balance fine-grained edge information and global context, incorporate modules that retain crucial high-frequency details through techniques such as wavelet transforms or frequency-domain processing, streamline the decoding stage with lighter attention operations, and introduce gradual multi-scale feature fusion strategies to minimize semantic gaps while keeping computation manageable.The rise of lightweight and edge-oriented Vision Transformers (AUHF-DETR) is helping narrow the accuracy efficiency gap between CNN-based and attention-based models, and ongoing advances in structured pruning and quantization-aware training are expected to reduce this gap even further.Improved training-time optimization. Methods such as SimOTA assignment, NMS-free training, balanced sample selection, and convergence-aware scheduling will become standard, as they enhance recall and robustness without adding to inference cost.Structured pruning is an effective approach for removing redundant layers, filters, or channels without significantly degrading performance. In addition, compact transformer modules can be integrated into lightweight backbones to further strengthen global feature modeling.Anchor-free detectors such as YOLOv8 and YOLOv11 offered for aerial scenarios due to their strong small-object sensitivity, but practical onboard deployment typically relies on their variants, which often need further optimization through pruning and quantization to meet real-time constraints.Knowledge distillation for large models Distillation from large vision or multimodal models is used to transfer semantic richness into smaller models. This improves robustness and fine-grained discrimination while keeping inference lightweight for design. The CLIP-KD algorithm introduces a more efficient version compared to the original CLIP model.Hardware-aware NAS integrated with compression. Neural Architecture Search will evolve toward pipelines that simultaneously account for quantization, pruning, operator fusion, and memory access patterns. This produces architectures that are inherently optimized for real-time edge execution rather than compressed post hoc.Advanced hardware-aware deployment. Quantization-aware training, structured and channel-level pruning, and TensorRT/NPU acceleration. These methods jointly reduce latency, model size, and energy consumption for onboard UAV processors.Cloud–edge tiling for high-resolution processing. High-resolution aerial and remote-sensing imagery will increasingly use tiling pipelines that split large frames into parallel patches. This enables cloud-side acceleration while the edge device maintains fast decision-making for navigation and safety-critical tasks.Federated edge learning for adaptive UAV perception.UAV systems will benefit from federated learning, updating models locally and sharing only compressed gradients. This reduces bandwidth usage, improves privacy, and allows adaptation to new environments without transmitting sensitive aerial data.Semantic enhancement via vision–language models. Integrating CLIP-based visual semantic embeddings and compressed multimodal backbones will improve cross-scene generalization and fine-grained recognition, even with limited aerial labels or domain shifts. It will need to integrate other compression techniques.Unified compression-first model design. Combine NAS, pruning, quantization, and reparameterization from the start, producing architectures that naturally satisfy latency, memory, and power constraints rather than requiring heavy post-processing.One-shot and weight-sharing architecture exploration. Future UAV detectors will increasingly rely on one-shot NAS and weight-sharing search spaces to rapidly evaluate large numbers of lightweight architectures. This avoids full training for each candidate and enables hardware-specific optimization for Jetson, NPU, or FPGA platforms.Integrating CLIP embedding with NAS and optimization (quantization or pruning) helps in reducing the size, with the potential of increasing the performance.Super-Resolution Branches Although not yet widely adopted, using a lightweight auxiliary super-resolution (SR) branch during training is emerging as a promising strategy for aerial small-object detection. The idea is to enhance high-resolution texture and edge information temporarily, lost during downsampling, without increasing inference cost, because the SR branch is removed during deployment (as in SuperYOLO). This approach improves feature quality for tiny objects while keeping the model efficient for onboard UAV hardware.

## 7. Conclusions

This survey goes beyond prior real-time studies by providing a tightly focused examination of onboard and edge-level aerial object detection. This survey emphasizes how modern lightweight design principles are applied across the entire detection pipeline. As feature extraction consumes more than 50% of computations compared to the remaining part of the object detector, it begins with lightweight convolutional feature extraction, showing how depthwise separable convolutions, partial convolutions, and efficient channel mixing reduce computation while preserving discriminative ability, which is an essential requirement for UAV platforms that operate under strict power and latency constraints. The discussion then moves to multiscale feature fusion in the neck, illustrating how streamlined FPN/PAN variants, cross-scale fusion blocks, and optimized aggregation modules generate rich spatial–semantic representations without introducing costly operations. Loss-function refinements, including adaptive IoU variants and task-aligned learning, are also analyzed for their role in stabilizing training and improving small-object sensitivity, which is often critical for aerial scenes.

The survey further investigates attention mechanisms, highlighting both lightweight fine-grained modules, such as channel or spatial attention, and soft pooling attention, and global modeling strategies inspired by Transformer designs. These modules enhance contextual understanding and long-range dependency modeling while being tailored for efficiency, making them viable for onboard processing. Building on this, the survey explores how vision–language models such as CLIP can be integrated into aerial detection pipelines to introduce semantic grounding, improve generalization to unseen categories, and support open-vocabulary detection. It describes strategies such as feature-space alignment, cross-modal distillation, or embedding projection so that CLIP’s strong semantic priors can be utilized without incurring the full computational overhead of large language models.

To address architectural exploration and hardware limitations simultaneously, the survey examines the role of Neural Architecture Search (NAS) in generating hardware-aware backbones and neck structures that match the resource budgets of UAV chips, edge GPUs, and FPGA fabrics. Complementary model compression techniques, such as structured pruning and mixed-precision quantization, are reviewed to demonstrate how they can reduce memory usage, execution latency, and energy consumption while preserving accuracy. The synthesis of these methods enables a path toward unifying lightweight design with larger pretrained models. CLIP or Transformer features can be selectively distilled into pruned, quantized, or NAS-optimized subnets that run efficiently on embedded GPUs or FPGA accelerators.

The survey also clarifies how hardware choice affects design decisions. GPUs typically deliver higher throughput and training flexibility, making them preferable for fog-node computation or heavier onboard processing when power budgets allow. FPGAs, although generally slower, offer significantly higher energy efficiency and deterministic low-latency execution, beneficial for endurance-critical UAV missions, long-flight platforms, or scenarios requiring tightly controlled timing. These distinctions motivate a forward-looking discussion of how emerging combinations of lightweight convolutional design, compact transformers, cross-modal embedding techniques, and hardware-aware NAS can drive progress toward accurate, fast, and resource-efficient aerial detection systems tailored for real-world onboard deployment.

## Figures and Tables

**Figure 1 sensors-25-07563-f001:**
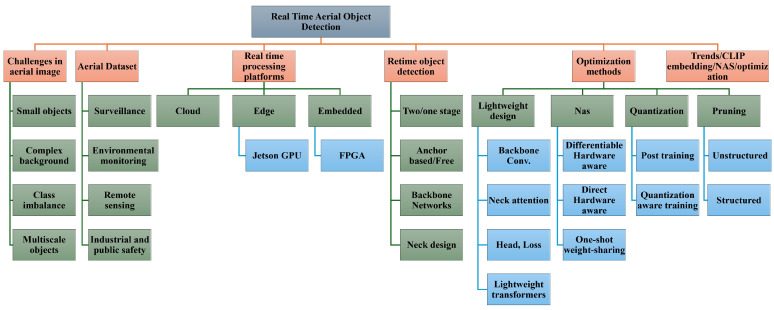
Overview of the survey’s main topics.

**Figure 2 sensors-25-07563-f002:**
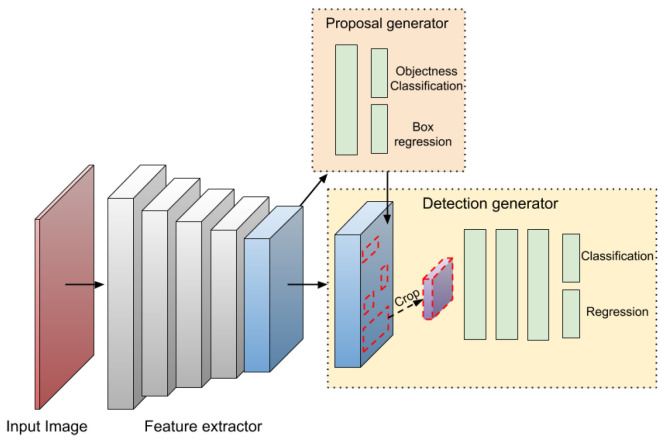
Two-stage algorithm main diagram [144].

**Figure 3 sensors-25-07563-f003:**
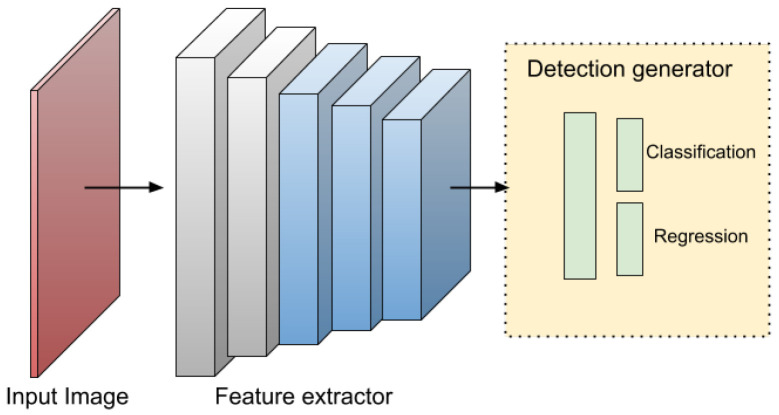
One-stage algorithm main diagram [144].

**Figure 4 sensors-25-07563-f004:**
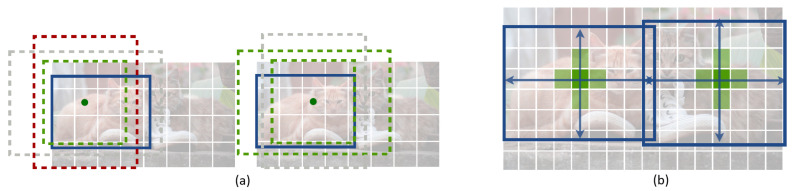
(**a**) Standard Anchor assignment, the green anchor denotes a positive match (IoU > 0.7), the red anchor indicates a negative sample (IoU < 0.3), and the gray anchor represents an ignored case with an intermediate IoU. The blue box shows the ground-truth object, (**b**) anchor-free center-based detection, the center pixel is treated as the positive location, surrounding pixels receive reduced negative loss, and the object dimensions are predicted through regression [145].

**Figure 5 sensors-25-07563-f005:**
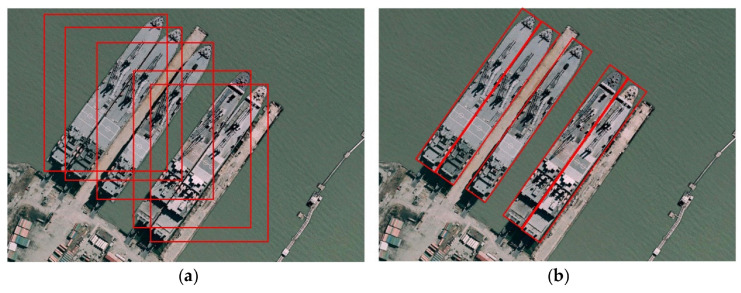
(**a**) Horizontal Boundary Boxes (HBB) and (**b**) Oriented Boundary Boxes (OBB) [150].

**Figure 6 sensors-25-07563-f006:**
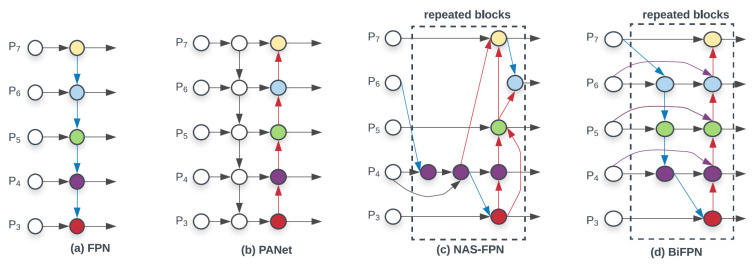
Different feature fusion networks [169].

**Table 1 sensors-25-07563-t001:** Summary of recent reviews and surveys on object detection.

Reference	Year	Main Contributions
[65]	2023	Reviews the evolution of object detection from early computer vision approaches in the 1990s to modern deep learning driven techniques, including milestone detectors, datasets, metrics, speedup strategies, and recent state-of-the-art methods.
[68]	2023	Provides details on algorithms designed for object-oriented aerial detection tasks.
[39]	2023	Reviews real-time processing algorithms, performance analysis, and sensor utilization for UAV-based object detection.
[9]	2024	Highlights challenges in aerial object detection and progress in deep learning algorithms addressing them, along with commonly used public aerial datasets.
[66]	2025	Classifies object detection methods into traditional and deep-learning paradigms, analyzing one-stage, two-stage, transformer-based, and lightweight models. Identifies gaps in lightweight research and compares algorithm performance using various metrics.
[67]	2025	Focuses on small object detection algorithms in aerial imagery, discussing techniques for improving precision and feature extraction.
[69]	2025	Focuses on real-time FPGA-based approaches and their performance.
This Survey		This survey examines the key challenges of aerial object detection and outlines lightweight design solutions across all stages of the detection pipeline for real-time and onboard design. It details quantization methods, hardware-aware NAS and its emerging design trends, and the growing use of CLIP-style visual semantic embeddings for fine-grained recognition and improved generalization to unseen aerial scenes. The discussion also covers optimization strategies for real-time deployment, efficient Transformer architectures, the associated design trade-offs, and practical model recommendations, tracing the evolution of lightweight detectors from 2020 to 2025.

**Table 3 sensors-25-07563-t003:** Real- time aerial studies tested on VisDrone2019/2021 with their corresponding performance and platforms used.

Ref.	Base Model	mAP50	Params (M)	Inference FPS/Latency ms	Platform
LODNU [47]	YOLOv4	31.4	8.7	68 FPS	NVIDIA RTX 3060
MFFSODNet [79]	YOLOv5	45.5	4.5	70 FPS	NVIDIA TITAN RTX
EA-YOLO [86]	YOLOv5s	39.9	5.9	N/A	NVIDIA RTX 3060
EL-YOLO [33]	YOLOv5s	N/A	7.6	29.4 FPS	NVIDIA RTX 3080 Ti
GCL-YOLO [39]	YOLOv5n	31.7	0.4	58 FPS	NVIDIA RTX 3060
	YOLOv5s	39.6	1.6	53 FPS	
	YOLOv5m	43.2	4.3	48 FPS	
	YOLOv5l	45.7	8.8	42 FPS	
AMMFN [12]	YOLOv5s	48.1	7.7	84.3 FPS	NVIDIA GTX 4090 Ti
EL-Net [45]	YOLOv7-tiny	38.7	2.0	N/A	NVIDIA RTX 3080
SOD-YOLO [46]	YOLOv7	50.7	30.3	72.5 FPS	NVIDIA RTX 4060
SOD-YOLO [123]	YOLOv8n	33.0	0.6	145 FPS	NVIDIA RTX 3090
	YOLOv8s	42.0	1.8	126 FPS	
[83]	YOLOv8s	31.0	6.0	128 FPS	NVIDIA RTX 4090
[82]	YOLOv8s	47.1	10.2	N/A	NVIDIA RTX 3090
LW-YOLOv8 [41]	YOLOv8m	42.3	13.4	72.3 FPS	NVIDIA Titan XP ×4
HRMamba-YOLO [84]	YOLOv8-m	N/A	33.5	31.0 ms	NVIDIA RTX 3090
MSFE-YOLO [89]	YOLOv8-s	41.4	N/A	101.0 FPS	GTX 3080 Ti
	YOLOv8n	33.8	N/A	149.3 FPS	
MFRENet [78]	YOLOv8m	53.4	27.1	50.5 FPS	NVIDIA RTX 2080

**Table 4 sensors-25-07563-t004:** Latency results in the two different deployment scenarios, edge and cloud [130].

Processing Environment	RTT Latency (ms)	Model Processing Latency (ms)	Communication Latency (ms)
Edge	35.09	32.59	2.50
Cloud	348.21	6.82	341.41

**Table 5 sensors-25-07563-t005:** Embedded/Edge Platforms with onboard aerial studies references.

Embedded/Edge Platform	Base Model	Reference
NVIDIA Jetson Xavier NX	YOLOv5	[50]
	YOLOv7	[115]
	YOLOv8	[92]
	YOLOv5	[51]
	YOLOv3	[57]
NVIDIA Jetson AGX Xavier	YOLOv8	[48]
	YOLOv7	[11]
	YOLOv5	[118]
NVIDIA Jetson TX2	YOLOv4	[49]
	YOLOv5	[115]
	YOLOv3	[56]
NVIDIA Jetson Nano	YOLOv7	[6]
	YOLOv4	[102]
	Mask R-CNN-ResNet18	[121]
	YOLOv8	[121]
	SSD-MobileNet	[121]
	SSD	[119]
NVIDIA Jetson AGX Orin	YOLOv7-Tiny	[100]
NVIDIA Jetson Orin Nano	YOLOv8	[35]
Huawei Atlas 200I DK A2	YOLOv5n	[46]
T710 NPU	YOLOv3	[5]
Raspberry Pi 3 B	SSD	[119]

**Table 6 sensors-25-07563-t006:** Summary of lightweight network architectures.

Network	Conv Type / Module	Key Features
SqueezeNet (2016) [157]	Fire module (1 × 1 conv replacing 3 × 3)	Reduced parameters and computation; flexible channel dimensions
MobileNetV1 (2017) [158]	Depthwise separable convolution	Efficient alternative to standard convolution; reduced computation
MobileNetV2 (2018) [159]	Inverted residual + pointwise conv	Expand-reduce feature maps; shortcut connections and linear activations
ShuffleNetV1 (2018) [160]	Pointwise group conv + channel shuffle	Bottleneck design; group-wise processing with shuffle to improve information flow
ShuffleNetV2 (2018) [161]	Split channels + conv + shuffle	Balanced computation; simplified structure; reduced element-wise ops
SqueezeNext (2018) [162]	Fire + depthwise separable conv	Improved parameter efficiency and model compactness
MobileNetV3 (2019) [163]	SE blocks + NAS + hard-swish	Enhanced efficiency and accuracy; optimized using neural architecture search
GhostNet (2020) [164]	Ghost modules with linear ops	Generate more features via cheap operations; reduce redundancy
FasterNet (2023) [165]	Partial convolution (PConv)+ PWConv	Low memory access and FLOPs; high throughput with efficient spatial feature extraction

**Table 7 sensors-25-07563-t007:** Comparison of Attention Mechanisms.

Attention Type	Year	Key Features	Limitations
SE (Squeeze-and-Excitation)	2018	Channel-wise attention using global average pooling and FC layers	Ignores spatial information; captures only channel dependencies
CBAM (Convolutional Block Attention Module)	2018	Sequential channel and spatial attention using convolutional operations	Limited to local context; ineffective for long-range dependencies
BAM (Bottleneck Attention Module)	2018	Parallel channel and spatial attention in a bottleneck structure	Convolution-based locality limits global context modeling
GCNet (Global Context Network)	2019	Global spatial attention via pooling and transform functions	Constrained by convolutional design; high computational load
ECA-Net (Efficient Channel Attention)	2020	Lightweight channel attention using 1D convolution without dimensionality reduction	Lacks spatial attention; only considers channel importance
SGE (Spatial Group-wise Enhance)	2020	Divides channels into groups for localized spatial attention; low overhead	Ignores inter-group channel dependencies; limited fusion
CA (Coordinate Attention)	2021	Positional encoding with efficient computation; suitable for mobile networks	Simplified spatial encoding may miss fine-grained relationships
NAM (Normalization-based Attention Module)	2021	Normalization-enhanced selection of important features	Underutilizes joint spatial-channel interactions
Triple Attention	2021	Simultaneous spatial and channel attention with low cost	Slightly higher complexity than lightweight modules
SA (Shuffle Attention)	2021	Integrates channel and spatial attention using grouped Shuffle Units	Needs explicit shuffling; moderate implementation overhead
SimAM (Simple Attention Module)	2021	Parameter-free 3D neuron-level attention via energy-based formulation	Simple structure; lacks tunable attention depth

**Table 8 sensors-25-07563-t008:** Summary of YOLO series models: architecture, features, and limitations.

Model (Year)	Anchors	Backbone	Neck	Head	Key Advancements/Limitations	Typical Use	Platform
YOLOv1 (2016) [183]	No anchors (direct regression)	Custom CNN (24 conv + 2 FC layers)	None	Fully connected detection head	Real-time speed with end-to-end training; limited small object detection and struggles with multiple objects per grid cell.	Real-time video streaming	Conventional and Desktop GPUs such as the Titan X GPU
YOLOv2 (2017) [184]	Anchor-based (predefined)	Darknet-19	None (passthrough layer)	Convolutional detection head	Introduced anchor boxes and batch normalization; improved accuracy; limited handling of overlapping objects.	Real-time video streaming	Conventional and Desktop GPUs, Titan X GPU
YOLOv3 (2018) [185]	Anchor-based (k-means clusters)	Darknet-53	FPN-like (multi-scale)	Convolutional detection head	Multi-scale prediction enhances small object detection; deeper backbone boosts accuracy but increases complexity.	Real-time video streaming, Industry. Yolov3 tiny is suitable for real-time onboard	Conventional, and Desktop GPUs. Only Yolov3 tiny is suitable for real-time onboard
YOLOv4 (2020) [146]	Anchor-based	CSPDarknet53	PAN + SPP	YOLOv3-style conv head	CSP backbone with Bag of Freebies and Specials; fast and accurate; complex training pipeline.	Real-time detection in production systems	Conventional and desktop GPUs such as (1080 Ti or 2080 Ti)
YOLOv5 (2020) [186,187]	Anchor-based (auto learning)	Modified CSPDarknet53	CSP-PAN	Decoupled head (cls, reg separate)	Deployment-focused modular design; widely adopted despite no official paper.	Powering security alarm systems or traffic monitoring, where low latency is non-negotiable. Some older NPU (Neural Processing Unit) drivers have highly optimized support specifically for the YOLOv5 architecture. YOLOv5n is suited for applications that demand extremely fast CPU inference with minimal latency.	Conventional GPUS Tesla T4 GPUs and desktop GPUs
YOLOX (2021) [188]	Anchor-free	CSPDarknet53 variant	PAN	Decoupled head	Anchor-free design simplifies training; decoupled head improves accuracy and convergence; uses SimOTA for label assignment.	Industrial applications	Conventional and desktop GPUs, YOLOXnano
YOLOv6 (2022) [189]	Anchor-free	EfficientRep (RepVG)	Rep-PAN	BiC-enhanced decoupled head	Self-distillation and Task Alignment Learning; Anchor-Aided Training (AAT) enhances lightweight accuracy without slowing inference.	Industrial applications in diverse scenarios, autonomous delivery robots	Conventional GPUS Tesla T4 GPUs and desktop GPUs
YOLOv7 (2022) [190]	Anchor-based	E-ELAN (Extended Efficient Layer Aggregation Network)	Extended PANet	Decoupled head with dynamic labels	Re-parameterization improves training; introduces coarse-to-fine label assignment and compound scaling.	General-purpose object detection	Conventional GPUs, V100 GPU and desktop GPus, YOLOv7-tiny is an edge GPU-oriented
YOLOv8 (2023) [187,191]	Anchor-free	CSPDarknet variant with CSPBottleneck and C2f module	enhanced FPN+PAN	Decoupled head	Improved modular design and contextual feature extraction; enhanced scalability across model sizes.	Keypoint detection (e.g., sports analytics), (small object detection). High-risk applications where every object must be detected, such as autonomous driving or security monitoring.	Conventional and desktop GPUs
YOLOv9 (2024) [192,193]	Anchor-free	Lightweight backbone with GELAN, CSPNet, RepConv	PAN-FPN	Decoupled head	Transformer modules enrich context; handles complex scenes effectively but with higher computation.	high-precision industrial inspection where false negatives are costly, small object detection in fields like satellite imagery or medical imaging, and complex scenes with occlusion or clutter that require preserving maximum feature information	Conventional and Desktop GPUs
YOLOv10 (2025) [193,194]	Anchor-free	Enhanced CSPNet; spatial-channel decoupled downsampling; large-kernel conv + partial self-attention	PANet	Dual head: one-to-many (training), one-to-one (inference)	NMS-free training and inference.	High-FPS video analysis, such as traffic or sports monitoring, and supports real-time robotics requiring low-latency navigation and obstacle avoidance	Conventional and desktop GPUs, NVIDIA 3090, YOLOv10n: Suitable for extremely resource-constrained environments
YOLOv11 (2025) [195]	Anchor-free	Enhanced backbone with C3k2 modules (a specification of GELAN)	PAN-FPN	Decoupled head	C3k2 improves gradient flow and semantics; 22% fewer parameters than YOLOv8m with higher mAP; versatile task support.	Real-time, high-accuracy tasks such as autonomous driving, edge-based detection, medical image analysis, and satellite imagery, where fast and precise object recognition is essential.	TensorRT10 FP16 on an NVIDIA T4 GPU,
YOLOv12 (2025) [196,197]	Anchor-free	Attention-centric backbone (FlashAttention)	PAN-FPN	Decoupled head with area attention	Attention-centric design with CNN-like speed; includes R-ELAN and position-aware attention; FlashAttention limits hardware compatibility.	Ideal for applications where precision is more important than real-time speed, medical imaging, quality control in manufacturing	FlashAttention is compatible only with GPUs from the Turing, Ampere, Ada Lovelace, or Hopper architectures, such as the T4, Quadro RTX series, RTX 20/30/40 series, RTX A5000/A6000, A30/A40, A100, and H100.
YOLO-GOLD (2025) [198]	Anchor-free	Enhanced CSPDarknet with grouped attention and NAS search	PAN-FPN + Selective Fusion	Lightweight hybrid head	Introduces Gather-and-Distribute (GD) mechanism for improved multi-scale fusion; combines NAS and attention for real-time UAV and embedded use; low-latency with competitive accuracy.	Target small and medium-size objects	Conventional GPUs, NVIDIA Tesla T4 GPU with TensorRT, and desktop GPUs.

**Table 9 sensors-25-07563-t009:** Developed aerial studies and their base detection model.

Base Models	References
YOLOv8	[17,35,41,42,43,44,48,52,61,62,78,81,82,83,84,87,89,90,92,94,95,96,97,98,104,105,116,121,123,200]
YOLOv5	[7,12,14,33,36,37,39,40,50,51,79,86,88,91,93,111,114,115,118,120]
YOLOv7	[6,11,34,45,46,100,110,201]
YOLOv4	[34,47,49,102,120,124,127]
YOLOv3	[5,56,57,202]
YOLOv10	[201]
YOLOv9	[201]
RT-DETR	[80,85]
SSD	[121]
R-CNN	[121]

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
