# Peer review of "Recent Real-Time Aerial Object Detection Approaches, Performance, Optimization, and Efficient Design Trends for Onboard Performance: A Survey"

_sensors, 2025, doi:10.3390/s25247563_

Round 1
Reviewer 1 Report
Comments and Suggestions for Authors
See the attachment.

Author Response
Thank you very much for your effort to review this manuscript. We appreciate your valuable comments. Please find the detailed responses below: (Kindly find the attached PDF file for responses)
|
Comments 1: Lines 24-59 list several major challenges facing aerial target detection, but lack specificity and systematic organization. For example, class imbalance is a long-standing, cross-disciplinary problem. The paper should focus on the characteristics of this problem in the field of aerial target detection. Furthermore, lines 51-59 and lines 24-31 can be merged. Considering that changes in illumination, background, and irrelevant targets are all background interference.
|
|
|
Comments 2: Section Two does not adequately explore the breadth of aerial datasets. Fine-grained object detection and recognition is a crucial task in aerial target detection. This content should be supplemented and explained in it.
|
|
Response 2: We have supplemented Section 2 to provide a more comprehensive overview of aerial datasets, with particular emphasis on fine-grained object detection and recognition. The revised section now includes detailed information, small object sizes and dimensions, as well as dataset characteristics such as platform types, and spatial resolutions. These additions provide a clearer understanding of the datasets’ suitability for fine-grained aerial target detection and recognition tasks in lines (274-306) More details added to Table2 page 9 for the datasets characteristic.
Comments 3: Sections 4 and 5 discuss algorithms that are too traditional and general. Tables 4, 5, and 6 all cover detectors for natural scenes, lacking information on target detectors for the aviation field. Specifically, there is excessive description of the YOLO series algorithms. The development of aviation target detectors and their components, as well as optimization designs, is rapid, with numerous related methods emerging. It is recommended that the article divide the relevant chapters into two parts: one for natural scenes and one for aviation scenes, introducing algorithms and optimizations for real-time target detectors.
Response 3: · To provide a clearer overview of object detection methods, we reorganized the discussion into two parts. The first part covers natural-scene detectors, summarizing conventional algorithms and lightweight networks in section 4 and it is divided to subsections for clarity sections 4.1, 4.2,4.3 and 4.4 (502 - 689) · The second part focuses on aerial aviation target detectors, highlighting algorithms and optimizations specifically designed for aerial imagery. (Section 4.5 line 691) Section 5 Optimization techniques line 752. · Lightweight backbone network (aerial detectors) grouped by design philosophy (efficiency through decomposition, channel operations, etc.) responding to another reviewer comment lines (764-922). · Table 4 is now renamed to Table 9 in page 22 The table lists aerial object detection models alongside the general base architectures from which they were developed. · Table 5 now renamed Table 8 page 21 for YOLO detectors. More information added for typical use and typical platform responding to another reviewer comment. The table includes details about the pipeline design to show later how each part of these detector is developed in the aerial network and to give insights which models suits aerial detection better. · Table 6 remain the same name on page 9, moved to section 4.2 with the general detectors. Showing the lightweight network used in general models and later in the light aerial networks discussion about how the aerial detectors developed and derived from the general lightweight network.
Comments 4: Section 6's analysis of design trends is overly simplistic and superficial, lacking a systematic summary. For example, optimizations of operators like attention mechanisms and convolution can be summarized as explorations of feature modeling. It needs to consider deeper directions, such as how to coexist with existing large vision-language model techniques.
Response 4: · Section 6 is now divided in to 3 subsections (Limitations in the current research, open challenge and Future directions. (Lines 1424-1563). · In section 5.1.1 and 5.12 description of design of backbones and neck structure with attention integrated in both details about how they improved to achieve lightweight models (773-848). · We have incorporated discussion on integration with large vision-language models (VLMs), including the use of CLIP embeddings in aerial detection. This addresses potential directions for lightweight model design, multimodal integration, and real-time processing. Added (section 5.6 In addition to recent studies that integrate NAS and LLM in section 5.7 )(lines 1326-1422)
Note: · Responding to another author comment, FPGA section 3.3 has been added (lines (469-481) with more information about embedded devices added in section 3.2 (lines 428,439) · Future direction include new trends for efficient RT-DETR (lines 1505-1515) and NAS (lines 1552-1560) more details are discussed for RT-DETR (lines 991-1076) and NAS (LINES 1176-1235) · Table 3 page 11 has been aged for more clarity about performance on Desktop GPUs for one dataset.
Thank you for taking the time to review our manuscript.
|
|
4. Response to Comments on the Quality of English Language |
|
Point 1: |
|
Response 1: (in red) |
|
5. Additional clarifications |
|
[Here, mention any other clarifications you would like to provide to the journal editor/reviewer.] |

Reviewer 2 Report
Comments and Suggestions for Authors
In regard to Section 5 Flow, it would be better for more effective organization to separate architectural design (Backbone, Neck, Head) from optimization techniques (Pruning, Quantization). In Consistent Naming, please standardize YOLO version identifiers (like YOLOv8n, YOLO-Nano) throughout the entire text. In Table 8, it is very important for proper model comparison to include quantitative analysis (i.e., mAP@0.5, FPS, Parameters (M)) in the Table 8 Performance Metrics. Expand Key Trade-offs, especially on the important input resolution vs. speed/accuracy trade-off for small object detection. Highlight New Trends, especially the new prominence and effectiveness of efficient Transformer-based models (RT-DETR) and Neural Architecture Search (NAS) for lightweight design.
Author Response
Thank you very much for your effort to review this manuscript. We appreciate your valuable comments. Please find the detailed responses below: (Kindly find the attached PDF file for the responses)
|
Comments 1: In regard to Section 5 Flow, it would be better for more effective organization to separate architectural design (Backbone, Neck, Head) from optimization techniques (Pruning, Quantization). In Consistent Naming, please standardize YOLO version identifiers (like YOLOv8n, YOLO-Nano) throughout the entire text. In Table 8, it is very important for proper model comparison to include quantitative analysis (i.e., mAP@0.5, FPS, Parameters (M)) in the Table 8 Performance Metrics. Expand Key Trade-offs, especially on the important input resolution vs. speed/accuracy trade-off for small object detection. Highlight New Trends, especially the new prominence and effectiveness of efficient Transformer-based models (RT-DETR) and Neural Architecture Search (NAS) for lightweight design."
|
· Section 6 is now divided in to 3 subsections (Limitations in the current research, open challenge and Future directions. (Lines 1424-1563). · Future direction include new trends for efficient RT-DETR (lines 1505-1515) and NAS (lines 1552-1560) more details are discussed for RT-DETR (lines 991-1076) and NAS (LINES 1176-1235) · Responding to other reviewer comment we have incorporated discussion on integration with large vision-language models (VLMs), including the use of CLIP embeddings in aerial detection. This addresses potential directions for lightweight model design, multimodal integration, and real-time processing. Added (section 5.6 In addition to recent studies that integrate NAS and LLM in section 5.7) (lines 1326-1422)
|
|
Note: · Responding to another author comment, FPGA section 3.3 has been added (lines (469-481) with more information about embedded devices added in section 3.2 (lines 428,439) · Table 3 page 11 has been aged for more clarity about performance on Desktop GPUs for one dataset.
|
|
Thank you for taking the time to review our manuscript.
|
|
4. Response to Comments on the Quality of English Language |
|
Point 1: |
|
Response 1: English language has been revised and improved. |
|
5. Additional clarifications |
|
[.] |

Reviewer 3 Report
Comments and Suggestions for Authors
This survey provides a comprehensive review of real-time aerial object detection approaches, with particular emphasis on onboard and edge deployment. The manuscript covers datasets, applications, processing platforms, algorithmic design strategies, and optimization methods for resource-constrained environments. While the scope is ambitious and the content is generally thorough, there are several areas requiring revision before publication.
Major Comments
Comment 1 (Lines 1-16, Abstract): The abstract needs significant revision for clarity and conciseness. The sentence structure is overly complex, particularly lines 3-9, which constitute a single run-on sentence. Consider breaking this into 2-3 shorter sentences. Additionally, the phrase "with less studies use" (line 8) is grammatically incorrect and should be revised to "with fewer studies using."
Comment 2 (Table 1, Page 4): The contribution of "This Survey" needs better differentiation from prior work. While you claim to focus on "onboard performance" and cover "models design progress from 2020 to 2025," references [19] and [47] appear to overlap significantly with your scope. Please explicitly clarify what unique contribution your survey makes beyond these existing reviews, particularly regarding the "in-depth analysis" you mention.
Comment 3 (Lines 296-313, Section 4): The discussion of anchor-based vs. anchor-free detection is well-presented, but the transition into this topic is abrupt. Consider adding 1-2 introductory sentences explaining why this distinction matters specifically for real-time aerial detection before diving into the technical details. Additionally, Figure 4 and Figure 5 should be referenced earlier in the text to improve flow.
Comment 4 (Table 8, Pages 25-27): This is one of the most valuable contributions of the survey, but it has critical formatting issues. Many cells contain "–" without clear indication of whether this means "same as baseline" or "not applicable." Please add a clear legend. Additionally, the "Opt. + Platform" column mixes optimization techniques with hardware specifications inconsistently. Consider splitting this into two separate columns for clarity.
Comment 5 (Lines 434-537, Section 5.1.1): The backbone network discussion provides excellent technical detail, but the organization could be improved. Consider restructuring chronologically (2016→2023) and grouping architectures by design philosophy (efficiency through decomposition, channel operations, etc.). Table 6 is helpful but appears late in the discussion—move it earlier and reference it consistently throughout this section.
Comment 6 (Table 3, Page 9): The performance metrics are inconsistently reported. Some studies report FPS, others report inference time (ms), and some report both. For a survey focused on real-time" performance, you should: (1) clearly define your threshold for "real-time" performance (currently mentioned as 30 FPS on line 228 but not consistently applied), and (2) normalize or convert all metrics to a common unit for fair comparison.
Comment 7 (Section 6, Lines 925-956): The "Design Trends" section reads more like a summary than an analysis of trends. What are the emerging directions? What gaps remain? Which approaches show the most promise for future onboard deployment? Consider restructuring this as "Findings and Future Directions" with subsections on: (1) Current state-of-the-art performance vs. requirements, (2) Trade-offs between different approaches, (3) Open challenges, and (4) Promising research directions.
Minor Comments
- Figure 1: The diagram is helpful but uses inconsistent terminology (e.g., "algorithms" vs. "Algorithm design"). Standardize terminology throughout.
- Lines 222-243: The discussion of processing platforms would benefit from a summary table comparing cloud, edge, and embedded approaches with their respective latency, accuracy, and cost trade-offs.
- Table 5: Excellent comprehensive comparison of YOLO versions, but consider adding a column for "typical use case" or "recommended platform" to make it more actionable.
Recommendation
Major Revision Required
This survey addresses an important and timely topic with significant practical implications. The breadth of coverage is impressive, and sections on lightweight design and model architectures are particularly strong. However, several critical issues must be addressed. I suggest major revision in this paper’s current form.

The English could be improved to more clearly express the research.
Author Response
Thank you very much for your effort to review this manuscript. We appreciate your valuable comments. Please find the detailed responses below: (Kindly find the attached PDF file for the responses)
|
Comments 1: (Lines 1-16, Abstract): The abstract needs significant revision for clarity and conciseness. The sentence structure is overly complex, particularly lines 3-9, which constitute a single run-on sentence. Consider breaking this into 2-3 shorter sentences. Additionally, the phrase "with less studies use" (line 8) is grammatically incorrect and should be revised to "with fewer studies using."
|
|
|
Comments 2: (Table 1, Page 4): The contribution of "This Survey" needs better differentiation from prior work. While you claim to focus on "onboard performance" and cover "models design progress from 2020 to 2025," references [19] and [47] appear to overlap significantly with your scope. Please explicitly clarify what unique contribution your survey makes beyond these existing reviews, particularly regarding the "in-depth analysis" you mention.
|
|
Response 2: · Table 1 has been modified and the contribution of this survey is modified as follows: (page 6) last raw “This survey examines the key challenges of aerial object detection and outlines lightweight design solutions across all stages of the detection pipeline for real-time and onboard design. It details quantization methods, hardware-aware NAS and its emerging design trends, and the growing use of CLIP-style visual semantic embeddings for fine-grained recognition and improved generalization to unseen aerial scenes. The discussion also covers optimization strategies for real-time deployment, efficient Transformer architectures, the associated design trade-offs, and practical model recommendations, tracing the evolution of lightweight detectors from 2020 to 2025.” · Responding to another reviewer comment, we have incorporated discussion on integration with large vision-language models (VLMs), including the use of CLIP embeddings in aerial detection. This addresses potential directions for lightweight model design, multimodal integration, and real-time processing. Added (section 5.6 In addition to recent studies that integrate NAS and LLM in section 5.7) (lines 1326-1422) · Responding to another reviewer, we add future details about efficient RT-DETR (lines 991-1076) and NAS (LINES 1176-1235) in aerial object detection.
Comments 3: (Lines 296-313, Section 4): The discussion of anchor-based vs. anchor-free detection is well-presented, but the transition into this topic is abrupt. Consider adding 1-2 introductory sentences explaining why this distinction matters specifically for real-time aerial detection before diving into the technical details. Additionally, Figure 4 and Figure 5 should be referenced earlier in the text to improve flow.
Response 3:
· Introductory sentences have been added (lines 519-526), details added to more clarify and for comprehensive and smoother explanation of these models (lines528-542) · To provide a clearer overview of object detection methods, we reorganized the discussion into two parts. The first part covers natural-scene detectors, summarizing conventional algorithms and lightweight networks in section 4 and it is divided to subsections for clarity sections 4.1, 4.2,4.3 and 4.4 (502 - 689) · The second part focuses on aerial aviation target detectors, highlighting algorithms and optimizations specifically designed for aerial imagery. (Section 4.5 line 691) Section 5 Optimization techniques line 752. · Lightweight backbone network (aerial detectors) grouped by design philosophy (efficiency through decomposition, channel operations, etc.) responding to another reviewer comment lines (764-922). · Figure 4 figure 5 have been referenced earlier in section 4.1 page 16 and 17 when general natural scene models have been discussed.
Comments 4: (Table 8, Pages 25-27): This is one of the most valuable contributions of the survey, but it has critical formatting issues. Many cells contain "–" without clear indication of whether this means "same as baseline" or "not applicable." Please add a clear legend. Additionally, the "Opt. + Platform" column mixes optimization techniques with hardware specifications inconsistently. Consider splitting this into two separate columns for clarity.
Response 4:
Comments 5: (Lines 434-537, Section 5.1.1): The backbone network discussion provides excellent technical detail, but the organization could be improved. Consider restructuring chronologically (2016→2023) and grouping architectures by design philosophy (efficiency through decomposition, channel operations, etc.). Table 6 is helpful but appears late in the discussion, move it earlier and reference it consistently throughout this section.
Response5: · To provide a clearer overview of object detection methods, we reorganized the discussion into two parts. The first part covers natural-scene detectors, summarizing conventional algorithms and lightweight networks in section 4 and it is divided to subsections for clarity sections 4.1, 4.2,4.3 and 4.4 (502 - 689) · The second part focuses on aerial aviation target detectors, highlighting algorithms and optimizations specifically designed for aerial imagery. (Section 4.5 line 691) Section 5 Optimization techniques line 752. · Lightweight backbone network (aerial detectors) grouped by design philosophy (efficiency through decomposition, channel operations, etc.) responding to another reviewer comment lines (764-922). · Table 6 has been moved to section 4.2 lightweight backbone (general networks) page 18
Comments 6: (Table 3, Page 9): The performance metrics are inconsistently reported.
Response 6:
Comments 7: (Section 6, Lines 925-956): The "Design Trends" section reads more like a summary than an analysis of trends. What are the emerging directions? What gaps remain? Which approaches show the most promise for future onboard deployment? Consider restructuring this as "Findings and Future Directions" with subsections on: (1) Current state-of-the-art performance vs. requirements, (2) Trade-offs between different approaches, (3) Open challenges, and (4) Promising research directions.
Response 7:
Minor coment1: Figure 1: The diagram is helpful but uses inconsistent terminology (e.g., "algorithms" vs. "Algorithm design"). Standardize terminology throughout. The diagram is changed and more structured.
Response MC1: · Figure 1. has been changed with a diagram that reflects the survey structure and topic.
Minor coment2: Lines 222-243: The discussion of processing platforms would benefit from a summary table comparing cloud, edge, and embedded approaches with their respective latency, accuracy, and cost trade-offs.
Response MC2: · We add Table 4 page 12, which includes comparison between edge and cloud with comparison line 422.
Minor coment3: Excellent comprehensive comparison of YOLO versions, but consider adding a column for "typical use case" or "recommended platform" to make it more actionable. Two columns have been added for typical use and platforms.
Response MC3:
Note: · Responding to another author comment, FPGA section 3.3 has been added (lines (469-481) with more information about embedded devices added in section 3.2 (lines 428,439) · Future direction include new trends for efficient RT-DETR (lines 1505-1515) and NAS (lines 1552-1560) more details are discussed for RT-DETR (lines 991-1076) and NAS (LINES 1176-1235) · Table 3 page 11 has been aged for more clarity about performance on Desktop GPUs for one dataset.
Thank you for taking the time to review our manuscript.
|
|
4. Response to Comments on the Quality of English Language |
|
Point 1: |
|
Response 1: English language has been revised and improved. |
|
5. Additional clarifications |
|
[Here, mention any other clarifications you would like to provide to the journal editor/reviewer.] |

Reviewer 4 Report
Comments and Suggestions for Authors
- The contributions and novelty have to be clearly highlighted. The authors have to be revised such as the revised version to be more clarified, compact and streamlined.
- The authors have to mention why this study has been conducted.
- The keywords are few and they have to be extended. Look at the words "real time" and "efficient"; where they are merely generic and abstract words.
- The introduction has weak literature review. There are little related works and I suggest to add more relevant works to enrich the article. FPGA-based lane-detection architecture for autonomous vehicles: A real-time design and development
- One can see that Table 1 has 7 works and there is no work within 2024; only 2023 and 2025.
- At the end of Introduction part, the authors have to state the contributions in points. One has to focus on the addition contributed by this study.
- The weakness of previous works in the literature has to be highlighted clearly. The motivation behind this review study has to be mentioned immediately after the literature reviewers.
- It is instructive to sort the literature based on types of aerial vehicles.
- The authors have to clarify how the article has been organized. There is vagueness and jumps in classifications.
- There are some figures which has been taken directly from previous works and they have low resolutions. The quality of figures have to be improved. See, for example, Figure 2 and 3.
- In the section "Optimization methods", the introduction of this part has not clarify the reasons behind optimizations and on which basis the optimization methods have been classified.
- The article descriptions based on block diagrams or schematic proposed methodologies of reviewed works.
- Table 8 is very important, but there is clear presentation of the table. The Table has to be revised.
- The authors have to present recommendations of this study. In addition, the limitations have also to be presented.
- The conclusion is descriptive. It is void of quantitative and numerical improvement and comparison.
- The future work has to be added and the references have to be updated.
- The organization and analysis of article have to be revised such that the article become more compact and streamlined.
Author Response
Thank you very much for your effort to review this manuscript. We appreciate your valuable comments. Please find the detailed responses below: (Kindly find the attached PDF file for the responses)
|
Comments 1: The contributions and novelty have to be clearly highlighted. The authors have to be revised such as the revised version to be more clarified, compact and streamlined.
|
|
Response 1: · Table 1 has been modified and the contribution of this survey is modified as follows: (page 6) last raw “This survey examines the key challenges of aerial object detection and outlines lightweight design solutions across all stages of the detection pipeline for real-time and onboard design. It details quantization methods, hardware-aware NAS and its emerging design trends, and the growing use of CLIP-style visual semantic embeddings for fine-grained recognition and improved generalization to unseen aerial scenes. The discussion also covers optimization strategies for real-time deployment, efficient Transformer architectures, the associated design trade-offs, and practical model recommendations, tracing the evolution of lightweight detectors from 2020 to 2025.” · Contribution added (lines 239-260) · Responding to another reviewer comment, we have incorporated discussion on integration with large vision-language models (VLMs), including the use of CLIP embeddings in aerial detection. This addresses potential directions for lightweight model design, multimodal integration, and real-time processing. Added (section 5.6 In addition to recent studies that integrate NAS and LLM in section 5.7) (lines 1326-1422) · Responding to another reviewer, we add future details about efficient RT-DETR (lines 991-1076) and NAS (LINES 1176-1235) in aerial object detection.
|
|
Comments 2: The authors have to mention why this study has been conducted.
|
|
Response 2: · Table 1 page 6 last raw as previous in addition to (lines211-237)
Comments 3: The keywords are few and they have to be extended. Look at the words "real time" and "efficient"; where they are merely generic and abstract words.
Response 3: · UAV, aerial, Object detection;edge ; real time; efficient; lightweight; onboard; optimization.
Comments 4: The introduction has weak literature review. There are little related works and I suggest to add more relevant works to enrich the article. FPGA-based lane-detection architecture for autonomous vehicles: A real-time design and development
Response 4: FPGA section 3.3 has been added (lines (469-481) with more information about embedded devices added in section 3.2 (lines 428,439)
Comments 5: One can see that Table 1 has 7 works and there is no work within 2024; only 2023 and 2025.
Response5: Thank you for pointing out this inconsistency. The survey on Page 6, Row 4 was mistakenly labeled as 2023; its correct publication year is 2024. We have corrected the year in Table 1 and verified all other survey dates to ensure accuracy.
Comments 6: At the end of Introduction part, the authors have to state the contributions in points. One has to focus on the addition contributed by this study.
Response 6: · The contributions list has been added, (lined 239-260) · Responding to another viewer comment, we have incorporated discussion on integration with large vision-language models (VLMs), including the use of CLIP embeddings in aerial detection. This addresses potential directions for lightweight model design, multimodal integration, and real-time processing. Added (section 5.6 In addition to recent studies that integrate NAS and LLM in section 5.7 )(lines 1326-1422)
Comments 7: The weakness of previous works in the literature has to be highlighted clearly. The motivation behind this review study has to be mentioned immediately after the literature reviewers.
Response 7: we have explicitly highlighted the limitations and gaps in existing review studies to clarify the weaknesses of prior work. Immediately following the literature review section, we added a dedicated paragraph outlining the motivation for our survey, explaining how our work addresses these gaps—particularly in terms of real-time aerial detection, hardware-aware optimization, and the integration of modern approaches such as efficient Transformers, NAS, and multimodal methods. These additions improve the logical flow and clearly position the contribution of our review within the context of prior studies. (lines 173-210)
Comments 8: It is instructive to sort the literature based on types of aerial vehicles.
Response 8:
As not all the research mentioned the platforms, we explore the datasets Platforms, and search about the types of platforms used in different aerial detection application and the dataset have been expanded and more information included about the image size resolutions small and fine grained, to give a big picture about the available datasets and their characteristic and challenges they address, small, fine grained (lines 299-306) “These datasets images are captured by different satellite and UAV platforms, which are commonly classified into three types; fixed-wing, rotary-wing, and hybrid designs. Fixed-wing drones operate at higher altitudes, cover large areas quickly, and offer long endurance due to their simple structure and efficient gliding capability, though they require runways for take-off. Rotary-wing UAVs can hover, take off, and land vertically, and fly at low altitudes to capture detailed, high-resolution data, making them well-suited for close-range inspection and sensing tasks. Hybrid platforms combine the strengths of both systems”
Comments 9: The authors have to clarify how the article has been organized. There is vagueness and jumps in classifications. Response 9: The structure of the survey has been detailed in lines (261-272).
Comments 10: There are some figures which has been taken directly from previous works and they have low resolutions. The quality of figures have to be improved. See, for example, Figure 2 and 3.
Response 10: Figures 2 and 3 have been updated with higher-resolution images to improve clarity.
Comments 11: In the section "Optimization methods", the introduction of this part has not clarify the reasons behind optimizations and on which basis the optimization methods have been classified.
Response 11: The introduction have been revisited and we clarify the reason behind optimization and on which basis they have been classified. (lines 753-763) Quoted below: “Aerial object detection systems deployed on UAV platforms face resource constraints in terms of computation, energy, and latency, making optimization a core requirement rather than an optional enhancement. Unlike ground-based detectors that can rely on powerful servers or desktop GPUs, onboard aerial systems must operate under limited processing power while still providing real-time and reliable detection. Therefore, optimization methods in the literature aim to either reduce the computational burden, enhance detection accuracy under resource limitations, or balance both goals to meet real-time performance requirements. Model compression for object detection is generally categorized into five main approaches: lightweight network design, pruning, quantization, knowledge distillation, neural architecture search (NAS), which are described in detail in the following subsections.”
Comments 12: The article descriptions based on block diagrams or schematic proposed methodologies of reviewed works.
Response 12: The article structure has been describe in Fig 1 page 7
Comments 13: Table 8 is very important, but there is clear presentation of the table. The Table has to be revised.
Response 13:
Comments 14: The authors have to present recommendations of this study. In addition, the limitations have also to be presented.
Response 14:
· We have incorporated discussion on integration with large vision-language models (VLMs), including the use of CLIP embeddings in aerial detection. This addresses potential directions for lightweight model design, multimodal integration, and real-time processing. Added (section 5.6 In addition to recent studies that integrate NAS and LLM in section 5.7 )(lines 1326-1422
Comments 15: The conclusion is descriptive. It is void of quantitative and numerical improvement and comparison.
Response 15: Conclusion has been revisited and updated.
Comments 16: The future work has to be added and the references have to be updated.
Response 16: Future directions have been added (Lines 1424-1563)., the references have been updated.
Comments 17: The organization and analysis of article have to be revised such that the article become more compact and streamlined.
Response 17: Addressing all reviewers comments which resulted in more clear and comprehensive structure.
Note: · Responding to another viewer comment future direction include new trends for efficient RT-DETR (lines 1505-1515) and NAS (lines 1552-1560) more details are discussed for RT-DETR (lines 991-1076) and NAS (LINES 1176-1235) · Table 3 page 11 has been aged for more clarity about performance on Desktop GPUs for one dataset.
Thank you for taking the time to review our manuscript.
|
|
4. Response to Comments on the Quality of English Language |
|
Point 1: |
|
Response 1: |
|
5. Additional clarifications |
|
[Here, mention any other clarifications you would like to provide to the journal editor/reviewer.] |

Round 2
Reviewer 1 Report
Comments and Suggestions for Authors
The problems have been reasonably resolved.
Reviewer 3 Report
Comments and Suggestions for Authors
The authors have addressed my comments.
Reviewer 4 Report
Comments and Suggestions for Authors
The authors have addressed all my concerns. The manuscript is considerably enhanced. Thank you.